# Ecological Restoration of Limestone Tailings in Arid Regions: A Synergistic Substrate–Plant Approach

**DOI:** 10.3390/biology15010082

**Published:** 2025-12-31

**Authors:** Wei Hou, Dunzhu Pubu, Duoji Bianba, Zeng Dan, Zengtao Jin, Qunzong Gama, Jingjing Hu, Yang Li, Zhuxin Mao

**Affiliations:** 1Forest Inventory and Planning Institute of Tibet Autonomous Region, Lhasa 850000, China; houwei80hou@163.com (W.H.); duojibianba@163.com (D.B.); zeng_dan2024@163.com (Z.D.); zengtaojin2021@163.com (Z.J.); yangzonggama@163.com (Q.G.); jingjinghu2018@163.com (J.H.); 2Xi’an Botanical Garden of Shaanxi Province, Institute of Botany of Shaanxi Province, Xi’an 710061, China; liyang@ms.xab.ac.cn (Y.L.); zhuxinmao@gmail.com (Z.M.)

**Keywords:** limestone tailings, arid regions, native grasses, *Pennisetum centrasiaticum*

## Abstract

Mining limestone in dry areas leaves behind piles of leftover rock and soil, called tailings, which damage the environment and need to be restored. We aimed to find the best way to help plants grow on these difficult sites in the arid regions of Northern China. We tested different mixes of natural soil with tailings and different water and nitrogen levels to see what would help native grass species grow best. We found that a mix of two parts soil to one part tailings, kept moderately moist, yielded the best results. Nutrient dynamics ultimately governed biomass accumulation, accounting for 57.8–84.2% of the biomass variation. Among the plants tested, *Pennisetum centrasiaticum* and *Setaria viridis* demonstrated the best overall performance, based on their comprehensive evaluation scores. This study shows that the key to restoring these areas is using the right soil mix, managing water carefully, and planting a smart combination of deep- and shallow-rooted grasses to use all available resources. These findings provide a practical, science-based guide for repairing damaged mining landscapes, which will help return life to these barren areas and improve the environment for local communities.

## 1. Introduction

The exploitation of limestone mineral resources in arid regions generates substantial volumes of alkaline tailings, i.e., fine-grained, calcium-carbonate-rich processing waste [1]. Unlike the parent rock, these tailings are typically loose, unstable, and characterized by high pH, a low organic matter content, and poor nutrient availability, particularly with respect to phosphorus and trace elements [1,2]. In arid landscapes, these material properties, combined with water scarcity and a high gravel content, create a substrate that is structurally unstable and biologically inhospitable, hindering natural revegetation and exposing surfaces to erosion [2,3]. Consequently, ecological restoration of such sites requires strategies that address both the inherent chemical constraints and the physical–hydrological limitations of the tailings material itself.

In situ soil reconstruction, which involves blending tailings with amendments to create a functional growth medium, presents a promising alternative to resource-intensive topsoil replacement [4,5,6]. However, the efficacy of this approach is highly substrate-specific. While successful applications have been documented for various types of tailings (coal, metal, etc.) [7,8,9,10,11,12], the unique geochemistry of limestone tailings—notably their high carbonate content and alkaline pH–fundamentally alters nutrient dynamics and plant–soil interactions. For instance, high calcium levels can induce phosphorus fixation and micronutrient imbalances [13]. Therefore, strategies developed for acidic or neutral tailings cannot be directly transferred, creating a critical knowledge gap regarding optimal amendment formulas and management practices for calcareous, arid tailings systems.

Vegetation establishment is the cornerstone of sustainable in-situ reconstruction [14,15]. Native species are particularly valuable owing to their pre-adaptation to regional climatic stresses such as drought [16,17]. A growing body of studies has demonstrated the outstanding contributions and capabilities of native plants in ecological restoration [18]. Yet, their tolerance to the specific edaphic stresses imposed by limestone tailings—high pH and low nutrient bioavailability—remains largely unquantified [16,19]. Key unknowns in this regard are whether native species can overcome these limitations under improved physical conditions (e.g., optimized tailings-to-soil ratios and moisture and nutrient regimes) and which functional traits confer success. Furthermore, the potential interactions between substrate modification and species selection are poorly understood, yet they are likely synergistic for successful restoration.

To address these interrelated gaps, we employed a controlled pot experiment using limestone tailings from an arid region in the eastern foothills of the Taihang Mountains in Northern China (38°15′ N, 114°32′ E). This area is characterized by a warm-temperate, semi-humid to semi-arid continental monsoon climate, with an average annual temperature of 8–14 °C and an annual precipitation of approximately 500 mm, concentrated predominantly in the summer months. The zonal soil type was limestone soil and some was shale soil [20]. The climatic conditions in this area, namely, significant evaporation and pronounced seasonal droughts further exacerbate the inherent challenges of tailings restoration, making effective water management and substrate optimization critical.

Based on this context, we hypothesized that: (i) plant biomass would be maximized under an optimal combination of tailings/soil ratio, water level, and nitrogen level; (ii) potassium (K) availability would be a primary nutrient limitation in this high-calcium system; and (iii) species with contrasting root architectures and resource-use strategies (e.g., deep-rooted perennial vs. shallow-rooted annual grasses) would exhibit differential performances. Our specific objectives were to (1) identify the optimal substrate ratio (tailings/soil) and moisture and nitrogen levels for plant establishment; (2) determine the key soil factors limiting plant growth, with a focus on nutrient dynamics; and (3) evaluate and rank the performance of five candidate native grass species to identify the most suitable species for restoring arid limestone tailings.

## 2. Materials and Methods

### 2.1. Experimental Materials

Limestone tailings samples were collected from three 10 m × 10 m plots located in the upstream, midstream, and downstream sections of the accumulation zones at the eastern foot of the Taihang Mountains (38°15′ N, 114°32′ E; elevation 410 m). Plots were selected based on their undisturbed conditions, considering both vegetation coverage and spatial representativeness. A diagonal sampling method was employed: five 0–20 cm sub-samples (collected from the four corners and the center of each plot) were thoroughly mixed to form a single 2 kg composite sample per plot, thereby reducing heterogeneity. On-site sieving using 0.5 mm and 10 mm stainless steel sieves ensured a particle size range of 0.5–1 cm, and any visible debris was manually removed.

Topsoil samples were collected from three 10 m × 10 m control plots, situated approximately 500 m from the mine boundary, where soil and topography were consistent with the regional background (the eastern foot of the Taihang Mountains—38°14′ N, 114°31′ E; elevation 390 m). The same diagonal sampling method was applied (five 0–20 cm sub-samples were mixed into one 2 kg composite sample per plot). All tailings and topsoil samples were transported to the laboratory within 24 h under cooled conditions (4 °C) and stored at 4 °C following air-drying and homogenization. The field-collected topsoil used as the planting medium had the following mean properties: pH—7.30; organic matter content—21.30 g/kg; total nitrogen content—1.92 g/kg; total phosphorus—0.86 content g/kg; total potassium content—25.03 g/kg; alkali-hydrolyzable nitrogen content—51.47 mg/kg; available phosphorus content—4.32 mg/kg; and available potassium content—148.45 mg/kg.

Seeds of five native plant species were collected near the limestone mine wastelands in 2020. Mature, healthy seeds were selectively harvested from at least 50 randomly chosen individual plants per species to capture genetic diversity. After collection, seeds were air-dried at room temperature, cleaned to remove debris, and stored in paper bags at 4 °C in the dark until the initiation of the experiment in 2023.

### 2.2. The Design of the Experiment

We employed a three-factor (substrate, moisture, and nitrogen) orthogonal L_9_(3^4^) array, generating nine treatment combinations (Table 1). Each treatment was replicated three times, resulting in 27 independent experimental units (pots).

The experiment was conducted in a greenhouse with pots arranged in a completely randomized design. The pots were cylindrical, with a volume of approximately 6 L (upper diameter, 26.5 cm; lower diameter, 15.5 cm; and height, 17.5 cm). Each pot contained one seedling to avoid intraspecific competition. The substrate factor varied tailings: soil ratios (2:1, 1:1, and 1:2 *w*/*w*). Soil moisture levels were maintained at 30%, 45%, and 60% of the maximum water-holding capacity (WHC) using a daily gravimetric method, where pots were weighed and replenished with deionized water until reaching their predetermined target weights to ensure precise moisture control. Nitrogen additions spanned 0, 200, and 400 g N·km^−2^ (as urea, equivalent to 0, 5, and 10 g N·m^−2^).

### 2.3. Greenhouse Conditions and Growth Management

The experiment was conducted from May to October 2023 in a rain-sheltered, open-sided greenhouse. This structure provided protection from uncontrolled rainfall but allowed for natural ventilation, such that the temperature, relative humidity, and photoperiod inside the greenhouse were consistent with the ambient conditions of the local Taihang Mountain foothills region. The seedlings were raised in a greenhouse. The potting matrix used for this nursery stage consisted of pure, unamended soil, which was identical to the soil component described in Section 2.1 and contained no tailings or other additives. After seedling emergence in June, uniform individuals were transplanted into the treatment pots.

Nitrogen treatments were applied in two split doses starting on 1 August. Soil moisture levels were maintained at 30%, 45%, and 60% of the soil’s maximum water-holding capacity (WHC). The procedure began on August 15 and was conducted daily throughout the experiment. Each pot was weighed using a digital balance. The amount of deionized water required to restore the pot to its predetermined target weight (corresponding to the specific WHC percentage for that treatment) was then calculated and added. This gravimetric method ensured precise and consistent moisture control across all replicates. Treatments were maintained until the harvest period. The total duration of the experiment was approximately 170 days.

### 2.4. Harvest and Sampling

Plants were harvested starting on 15 October. For each of the 27 experimental pots (biological replicates), the following procedure was conducted: Plant height was recorded. The entire aboveground biomass and the complete root system were carefully separated. These constituted one plant sample per pot. Concurrently, soil was collected from the root zone of the same pot. From each pot, three spatially distributed soil subsamples were taken and thoroughly homogenized to form a single composite soil sample per pot. Thus, each pot yielded one composite soil sample and one plant sample for subsequent analysis. All aboveground and belowground plant materials were oven-dried at 65 °C until reaching constant weight for biomass determination.

### 2.5. Soil Physicochemical Analysis

All soil analyses were performed on the composite samples (one per pot, with n = 27). Soil analyses were performed concurrently. Soil pH was measured in a 1:2.5 (*w*/*v*) soil-water suspension. Total nitrogen (TN) was quantified using the Kjeldahl digestion method [21]. Total phosphorus (TP) was determined using sodium hydroxide fusion followed by molybdenum antimony blue colorimetry, while available phosphorus (AP) was extracted with 0.5 M NaHCO_3_ (pH 8.5) and measured using the molybdenum blue colorimetry method [22]. Total potassium (TK) and available potassium (AK) concentrations were quantified using flame photometry [23]. Alkali-hydrolyzable nitrogen (AN) content was determined using the alkaline diffusion method [24].

### 2.6. Statistical Analyses

Raw data were cleaned and organized in Microsoft Excel prior to analysis. Orthogonal range analysis was used to assess the effect magnitude of each factor and identify preliminary optimal levels. The effect magnitudes of each factor were quantified using range values (R):(1)$$R=\mathrm{Max}\left(K_{i}\right)-Min(K_{i})$$
where *K_i_* represents the mean response value at level *i* of the factor. The optimal level for each factor was identified as the highest *K_i_* value.

To statistically compare biomass outcomes across the nine specific treatment combinations, the data were also subjected to a one-way ANOVA considering ‘Treatment Combination’ as a fixed factor with nine levels, followed by Tukey’s HSD post hoc test for multiple comparisons (α = 0.05).

Given the orthogonal array design (not a full factorial design), a three-way ANOVA model appropriate for unbalanced designs was applied. We acknowledge that the L_9_ design does not allow the estimation of all two- and three-way interactions independently. Therefore, in the ANOVA, we focused on testing the main effects of substrate (A), moisture (B), and nitrogen (C), with the model residuals encompassing interactions that were not estimated. This approach is valid for screening main effects in orthogonal designs. ANOVA was performed with all species pooled, followed by species-specific analyses. Post hoc comparisons were made using Tukey’s HSD test with FDR correction (α = 0.05).

Stepwise multiple linear regression (entry α = 0.05, removal α = 0.10) was used to identify key soil predictors of biomass. Multicollinearity was assessed using variance inflation factors (VIF < 10). Structural equation modeling (SEM) was then employed to test hypothesized causal pathways. Model fit was evaluated using multiple indices: *p* > 0, 1 < χ^2^/df < 3, Comparative Fit Index (CFI) > 0.90. Models were trimmed iteratively based on modification indices and theoretical justification.

Principal component analysis (PCA) was performed on a correlation matrix of standardized biomass and soil variables to reduce dimensionality. Species were ranked based on composite scores from components with eigenvalues > 1, weighted by explained variance.

SAS version 8.1 (SAS Institute Inc., Cary, NC, USA), was employed for three-way ANOVA, Tukey’s multiple comparisons, and multiple linear regression; PCA was conducted using SPSS Statistics 26.0 (IBM Corp., Armonk, NY, USA); and SEM was implemented in AMOS 26.0 (IBM SPSS, Armonk, NY, USA) using maximum likelihood estimation. All figures were prepared using GraphPad Prism 8 and PowerPoint.

## 3. Results

### 3.1. Plant Biomass Responses to Substrate, Water, and Fertilization Treatments

Orthogonal range analysis and multiple comparisons revealed distinct optimal treatments across species (Table 2). *E. indica*, *P. centrasiaticum*, *A. splendens*, and *L. chinensis* attained maximal biomass under the T_3_ treatment (*p* < 0.05). In contrast, *S. viridis* reached peak biomass in T_5_, though its aboveground biomass (AGB) and total biomass (TB) did not differ significantly between T_3_ and T_5_ (*p* > 0.05). Range analysis further demonstrated that substrate composition had the strongest effects on plant responses, followed by soil moisture regime, with nitrogen fertilization showing the least influence.

Three-way ANOVA revealed significant main effects of substrate and soil moisture on AGB, belowground biomass (BGB), and TB (all *p* < 0.05, Figure 1), but no significant effects on biomass allocation (root: shoot ratio, all *p* > 0.05, Figure 1).

All species exhibited a significant reduction in biomass with increasing tailings proportion (Figure 1). Species sensitivity varied substantially, forming a clear tolerance gradient. *P*. *centrasiaticum* proved to be the most tolerant, with total biomass (TB) under high tailings reduced by only 35.66% relative to the low-tailings control. In contrast, *L*. *chinensis* was the most severely affected, with TB declining by 84.14%. *A*. *splendens* also showed high sensitivity (TB reduced by 73.05%), while *S. viridis* and *E. indica* displayed intermediate reductions (TB reduced by 59.22% and 52.23%, respectively).

Conversely, increased soil moisture significantly enhanced biomass accumulation across all species (Figure 1). *E*. *indica* showed the most dramatic positive response, with TB increasing by 200.75% under high moisture conditions. *A. splendens* and *P. centrasiaticum* also exhibited strong, moisture-dependent growth, with TB increasing by 150.35% and 88.15%, respectively. *S. viridis* and *L. chinensis* benefited to a more moderate extent (TB increased by 56.12% and 70.21%, respectively).

### 3.2. Soil pH Responses

Three-way ANOVA revealed significant interactive effects of substrate, soil moisture, and fertilization on soil pH (*p* < 0.05, Figure 2), with the main effects varying among plant species. For *E. indica*, increasing the tailings proportion from the low level resulted in a significant pH increase, namely, by 0.14 units at the medium level and by 0.17 units at the high level. For *S. viridis* and *L. chinensis*, a medium tailings proportion (as opposed to a low level) significantly decreased pH by 0.15 and 0.11 units, respectively. However, with a high tailings proportion (as opposed to a low level), pH significantly increased for *P. centrasiaticum*, *A. splendens*, and *L. chinensis* (by 0.14, 0.06, and 0.36 units, respectively), indicating a potential non-linear or species-specific response threshold. Elevated soil moisture levels (as opposed to a low moisture level) consistently exerted a positive main effect on soil pH. The increases were most pronounced in *P. centrasiaticum* (increasing by 0.10 units under high-moisture-level conditions) and *A. splendens* (increasing by 0.06 units). A medium nitrogen application rate (as opposed to no application) significantly raised pH for *E. indica* (by 0.10 units) and *A. splendens* (by 0.05 units) but lowered it for *P. centrasiaticum* and *L. chinensis* (by 0.10 and 0.11 units, respectively). A high nitrogen rate had an acidifying main effect, further reducing pH in *P. centrasiaticum* and *L. chinensis* (by 0.11 and 0.13 units, respectively, relative to no application).

### 3.3. Soil Nutrient Dynamics Responses

#### 3.3.1. Changes in Total Soil Nutrients

Three-way ANOVA indicated that substrate and nitrogen fertilization significantly increased soil total nitrogen content (TN) (both *p* < 0.05, Figure 3). Medium and high tailings proportions increased TN content by 11.13% and 13.81%, respectively, while medium and high levels of nitrogen application raised TN content by 14.27% and 16.68%, respectively. Species-specific responses varied significantly (Figure 3). The *E. indica*-cultivated soil exhibited the strongest substrate effect, with TN increasing by 20.54% under the high tailings proportion. In contrast, TN content was not influenced by the substrate in either the *P. centrasiaticum*- or *A. splendens*-cultivated soil. Only the TN content of the *P. centrasiaticum*-cultivated soil was influenced by soil moisture, with high moisture levels increasing it by 28.14%. The *A. splendens*-cultivated soil exhibited the greatest sensitivity to nitrogen fertilization, showing a 30.12% increase in TN content under medium nitrogen application.

Substrate composition significantly influenced total phosphorus (TP) content. Medium and high tailings proportions significantly reduced TP content by 3.76% and 5.45%, respectively (*p* < 0.05, Figure 4). All the plant–soil systems showed a decrease in TP content with an increase in the tailings proportion, with the greatest reduction occurring in *E. indica*-cultivated soil (8.17–9.94%) and the smallest in *P. centrasiaticum*-cultivated soil (1.62–4.06%). High soil moisture levels significantly increased the TP content of the *S. viridis*- and *E. indica*-cultivated soil by 2.74% and 2.07%, while high levels of nitrogen fertilization notably decreased the TP of the *P. centrasiaticum*- and *L. chinensis*-cultivated soil by 1.64% and 1.85%, respectively.

Total potassium (TK) content was also affected by the type of substrate. Medium and high tailings proportions significantly reduced TK content by 2.65% and 3.33%, respectively (*p* < 0.05, Figure 5). All the plant–soil systems exhibited a decrease in TK content with an increase in tailings proportions. The *P. centrasiaticum*-cultivated soil exhibited the smallest reduction, with no significant changes under the medium tailings proportion and a 2.89% reduction under the high tailings proportion. The *E. indica*-cultivated soil exhibited the most sensitive reduction, with a 4.48% decrease under the medium tailings proportion. Only medium levels of water addition increased the TK content of the *E. indica*-cultivated soil, specifically by 4.16%, whereas nitrogen fertilization reduced the TK content of the *E. indica*-cultivated soil by 4.06% and 3.54%.

#### 3.3.2. Changes in Soil Available Nutrients

Substrate, soil moisture, and nitrogen fertilization all significantly influenced soil available nitrogen (AN) (all *p* < 0.05, Figure 6). Medium and high tailings proportions increased AN by 19.57% and 54.12%, respectively, while nitrogen fertilization increased AN by 22.28% and 50.20%, respectively. In contrast, medium and high soil moisture levels reduced AN by 28.00% and 25.17%, respectively. Species-specific responses varied considerably (Figure 4). The *E. indica*-cultivated soil exhibited the largest AN increase (89.42%) under a high tailings proportion, while the *P. centrasiaticum*-cultivated soil exhibited the smallest increase (34.62%). The *S. viridis*-cultivated soil showed the greatest sensitivity to moisture addition, with AN decreasing by 40.58% under medium moisture conditions. The *S. viridis*-cultivated soil displayed exceptional responsiveness to medium levels of nitrogen addition, with a 63.89% increase, whereas the *E. indica*-cultivated soil showed the greatest increase (83.75%) under high levels of nitrogen addition.

Medium and high soil moisture levels decreased available phosphorus (AP) by 13.88% and 14.19%, respectively, across all treatments (*p* < 0.05, Figure 7). Under a medium tailings proportion, the AP in the *S. viridis*- and *L. chinensis*-cultivated soil increased significantly by 22.18% and 37.31%, respectively, while the AP in the *P. centrasiaticum*-cultivated soil decreased by 15.57%. Water application decreased AP by 17.57–23.85% across species, with the maximum reduction observed in the *S. viridis*-cultivated soil (23.85%).

High tailings proportions increased available potassium (AK) by 4.63% overall (*p* < 0.05, Figure 8). AK in the *E. indica*- and *P. centrasiaticum*-cultivated soil was only affected by medium soil moisture levels, decreasing by 7.40% and 4.20%, respectively. AK in the *S. viridis*-cultivated soil responded strongly to nitrogen addition, increasing by 7.51% and 8.76% under medium and high levels of nitrogen addition, respectively. The AK in the *A. splendens*- and *L. chinensis*-cultivated soil increased by 3.10% and 5.27% under medium tailings proportions, and by 6.24% and 8.35% under high tailings proportions, respectively.

### 3.4. Relationships Between Plant Biomass and Soil Physicochemical Properties

Stepwise regression analysis revealed significant correlations between soil potassium content (both total potassium and available potassium) and biomass parameters (aboveground biomass, belowground biomass, and total biomass) across all five plant species (all *p* < 0.05, Table 3). Specifically, soil total potassium content (TK) showed significant positive correlations with plant biomass (Standardized β: 0.397–0.603), while soil available potassium content (AK) exhibited significant negative correlations with biomass (Standardized β: −0.825–−0.391). Moreover, *S. viridis* and *A. splendens* demonstrated significant negative correlations between aboveground/total biomass and soil available phosphorus (AP). Soil pH was observed to have significant effects on the biomass or below-ground biomass of *L. chinensis* and *S. viridis*.

The SEM analysis revealed distinct treatment-mediated pathways across species (Figure 9). In *E. indica*, treatments exerted a positive effect on available potassium (AK, β = 0.36, *p* < 0.10) that subsequently suppressed biomass accumulation (β = −0.81, *p* < 0.001), accounting for 70.3% of the variance (Figure 9a). *S. viridis* demonstrated dual mechanisms: treatments not only reduced total potassium content (TK, β= −0.65, *p* < 0.001), but also directly decreased biomass production (β= −0.50, *p* < 0.05) through sequentially influenced TK and AK, with the combined pathways explaining 60.3% of the variation. It is important to note that the standardized indirect effect of TK content on biomass through this pathway was +0.21 (Figure 9b). More complex interactions emerged in *P. centrasiaticum*, where treatments significantly lowered soil pH (β = −0.49, *p* < 0.05) and TK content (β = −0.36, *p* < 0.05). The acidified soil marginally strongly reducing ammonium nitrogen levels (AN; β = −0.67, *p* < 0.001), with all three parameters directly influencing biomass (β = 0.37, *p* < 0.05; β = 0.50, *p* < 0.05; and β = 0.58, *p* < 0.001, respectively), explaining 65.7% of the variance (Figure 9c). The treatment effects in *A. splendens* operated primarily through pH-mediated available phosphorus (AP) reduction (pH → AP: β = −0.52, *p* < 0.05; AP → biomass: β = −0.44, *p* < 0.05), combined with direct negative treatment impacts (β = −0.51, *p* < 0.05), accounting for 57.8% of the variance (Figure 9d). Most notably, *L. chinensis* exhibited the strongest responses, with both direct biomass suppression (β = −0.52, *p* < 0.001) and multi-step pH-TN-AN/AP/AK regulation, ultimately explaining 84.2% of the biomass variation (Figure 9e).

### 3.5. Optimal Plant Selection

Principal component analysis (PCA) was performed on a correlation matrix (n = 135) derived from standardized plant growth and soil response variables (mean = 0, and standard deviation = 1) to eliminate scale differences. The analysis yielded four principal components (PCs) with eigenvalues > 1, explaining 31.16%, 18.58%, 15.54%, and 9.94% of the total variance, respectively, with a cumulative variance of 75.21% (Table 4).

To integrate the multidimensional information into a single metric of overall performance, a comprehensive evaluation score (*F*) was calculated for each species using the following weighted summation formula:$$F_{i}=\sum {PC}_{ij}\times W_{j}$$
where *F_i_* is the comprehensive score for the *i*-th species, *PC_ij_* is the score of the *i*-th species on the *j*-th principal component, and *W_j_* is the weight of the *j*-th component, defined as its variance contribution rate (i.e., *W_j_* = Eigenvalue*_j_*/Total Variance).

Thus, the calculation for each species was: *F* = (PC1 score × 0.3116) + (PC2 score × 0.1858) + (PC3 score × 0.1554) + (PC4 score × 0.0994). The species were then ranked in descending order with respect to their F scores, indicating a decreasing overall restoration efficacy (Table 5): *P. centrasiaticum* (highest score), followed by *S. viridis*, *L. chinensis*, *A. splendens*, and *E. indica*.

## 4. Discussion

This study systematically evaluates ecological restoration strategies for arid limestone tailings via a controlled pot experiment, integrating substrate optimization with native plant selection. The key findings demonstrated that (1) an optimal substrate mixture containing approximately 30% tailings, combined with precise moisture management (60% of WHC), maximized initial plant establishment. (2) Soil potassium dynamics served as a key indicator of plant–soil feedbacks, where the negative correlation between available potassium and biomass primarily reflected nutrient depletion due to plant uptake under the closed experimental system, rather than potassium being the ultimate growth-limiting factor. (3) The C_4_ grasses *P*. *centrasiaticum* and *S*. *viridis* were identified as the most suitable native species, exhibiting complementary functional traits. Based on these results, a practical restoration framework is proposed, emphasizing the need for integrated management of substrate structure, water, and key nutrients. These findings provide mechanistic insights into the early-stage restoration of calcareous tailings under arid conditions, while highlighting the necessity for future field validation to assess long-term species interactions and ecosystem development.

### 4.1. Substrate and Water–Nutrient Regulation

Our results identify substrate composition as the primary factor controlling plant growth in arid limestone tailings. Optimal growth was achieved with a 2:1 soil-to-tailings mixture, which improved substrate structure by increasing porosity and reducing compaction, thereby creating a more favorable environment for plant establishment [8,11,25]. This finding aligns with earlier studies on rocky desertification that suggested excessive soil amendment did not necessarily improve restoration outcomes [26]. Conversely, the direct application of large quantities of untreated soil can increase compaction [27] and inhibit plant growth [14,28]. Our results confirm that incorporating a controlled proportion of tailings can alleviate such adverse effects, supporting the view that tailored substrate design is crucial for effective phytoremediation.

Soil moisture emerged as another critical limiting factor. Increased moisture partially counteracted the tailings’ inhibitory effects on plant biomass, highlighting moisture’s essential role in arid tailings restoration. Water scarcity directly impairs key physiological processes such as photosynthesis, respiration, and stomatal regulation, leading to reduced biomass accumulation and plant survival [29,30,31]. Thus, water availability constitutes a definitive constraint for ecological restoration in arid limestone tailings systems.

In contrast, relative to substrate type and water, nitrogen fertilization exhibited the most limited influence. Range analysis revealed it had the smallest factor effect, and ANOVA confirmed that nitrogen application significantly affected only *L. chinensis* biomass, with no detectable impact on the other four species. These results support previous reports that inorganic amendments such as urea addition may be ineffective in enhancing long-term substrate quality and native plant growth in mine waste rehabilitation [8].

The muted response to nitrogen fertilization can be attributed to the inherently sufficient nitrogen content in the topsoil used for amendment. This notion is supported by the measured total soil nitrogen (STN) content, 1.92 g/kg, which is notably higher than the typical range for farmland soils in Northern China (approximately 0.50–0.81 g/kg) [32]. Hence, nitrogen was not the primary limiting nutrient in this system, explaining the plants’ subdued responses to fertilization.

In summary, our findings establish a clear hierarchy of the factors influencing the ecological restoration of arid limestone tailings: substrate composition acted as the primary driver, followed by the critical constraint of water availability, while nitrogen fertilization played a comparatively minor role due to the inherent sufficiency of nitrogen in the tailings matrix.

### 4.2. Shifts in Potassium Pool Equilibrium Driven by Plant–Soil Interactions

Soil potassium derives mainly from the weathering of potassium-bearing minerals (e.g., feldspars and micas) and occurs in three functionally distinct pools: readily available (soil-soluble and exchangeable), slowly available (non-exchangeable), and structural (mineral-lattice-bound) forms [33,34,35]. These pools maintain a dynamic equilibrium [36]. Our results reveal a significant positive correlation between soil total potassium (TK) content and plant biomass, alongside a negative correlation between available potassium (AK) and biomass. It is important to note that, in our study, the available nutrients primarily reflected the post-uptake nutrient status, serving as an outcome of plant growth productivity rather than an indicator of initial availability. Rather than indicating that potassium itself is the ultimate growth-limiting factor, this interpretation, which does not contradict the fundamental role of potassium in plant physiology, suggests that the observed pattern likely resulted from shifts in plant demand and alterations in soil processes under varying growth conditions.

In treatments with a high proportion of tailings, plant biomass decreased by 35.66% to 84.14%. This reduction in growth likely lowered plant potassium uptake, allowing available potassium to accumulate in the soil. For instance, in this study, available potassium in the soil containing *L. chinensis*, *S. viridis*, and *A. splendens* increased by 8.35%, 3.99%, and 6.24%, respectively. Simultaneously, the incorporation of tailings, with their distinct mineralogy and pH, may have perturbed the equilibrium between potassium pools [37,38,39]. This could have increased the release of non-exchangeable potassium into the available pool through abiotic processes, such as accelerated mineral weathering within the amended soil matrix [36]. Furthermore, an improved soil microenvironment might also stimulate potassium-solubilizing bacteria [37], which can mineralize structural potassium, thereby depleting the total potassium pool while increasing the available fraction [40]. Therefore, the total potassium content in the soil subjected to high-tailings treatments decreased significantly by 2.89% to 3.70% in this study. However, as we did not directly measure microbial activity or mineral weathering rates, the proposed involvement of potassium-solubilizing bacteria remains speculative and requires verification.

Conversely, in the treatments with optimal substrate and moisture conditions that promoted biomass, enhanced plant uptake drew down the available potassium pool, resulting in the observed negative correlation. The positive correlation with total potassium likely reflects the greater inherent potassium content in treatments with more soil, a factor that also supported higher biomass production. Therefore, the observed potassium dynamics appear to be a consequence of plant growth responses to substrate and water availability, not their primary driver. The critical role of plant uptake in regulating soil potassium pools was further evidenced by the response to water supplementation. Increases in biomass production in *E. indica*, *P. centrasiaticum*, and *A. splendens*, by 200.75%, 88.15%, and 150.35%, respectively, led to potassium sequestration in plant tissues and consequent 1.43%, 1.05% and 2.78% decreases in soil available potassium. This growth-driven immobilization process effectively depleted the available potassium pool.

In summary, the positive correlation between biomass and total potassium reflects the standing stock of this nutrient, while the inverse relationship with available potassium indicates its active biological consumption. Together, these relationships demonstrate that potassium dynamics in this system are governed by the interplay between plant demand and modifications of soil processes.

### 4.3. Optimal Plant Selection

Principal component analysis and comprehensive evaluation identified *P. centrasiaticum* as the best-performing species for restoring arid limestone mine tailings, demonstrating remarkable ecological adaptability. As a C_4_ plant, *P. centrasiaticum* maintains high photosynthetic efficiency under environmental stress [41,42], explaining the relatively minimal biomass reduction (35.66%) observed at the highest tailings proportion in our study. This result aligns with its documented resilience against extreme conditions [43]. This species’ stress adaptation involves integrated physiological and structural modifications. Physiologically, it can employ coordinated photoprotection through reversible downregulation of PSII photochemistry and increased thermal energy dissipation [43], combined with the C_4_ biochemical pathway [44] to maintain efficient carbon assimilation and water use efficiency under stress. Structurally, *P. centrasiaticum* develops an extensive root system [45]. In this study, its root biomass exceeded that of the other four species. This robust root architecture enables effective acquisition of deep soil water and nutrients [46], promotes rapid clonal expansion, and enhances ramet survival through strong clonal integration [47,48,49]. Owing to its exceptional adaptation to harsh environments, *P. centrasiaticum* has become a key native species for ecological restoration in China’s arid regions.

Our results, together with findings reported in previous studies, confirm that *S. viridis*, another C_4_ species, can serve as an ideal pioneer plant in restoration initiatives [3,50,51]. It functioned as a complementary secondary species in our model for three main reasons. First, *S. viridis* produces long-lived seeds and exhibits rapid early growth, enabling quick ground cover that mitigates rain-drop erosion [52]. Second, although its shallow roots limit access to deep nutrients, the species can enhance surface phosphorus availability through its high phosphatase activity and phosphate-solubilizing microbial communities [53]. Third, previous studies indicate that *S. viridis* can adapt to tailings [54] and decertified soil [55] by regulating its antioxidant system and modulating heavy-metal accumulation. Moreover, it can improve the soil microenvironment by recruiting stress-tolerant microorganisms [56], thereby facilitating the establishment of perennial species like *P. centrasiaticum*.

Based on these findings, we propose a restoration model combining *P. centrasiaticum* and *S. viridis*. This approach aligns with earlier work showing that multi-species planting in tailings rehabilitation enhances plant survival [8] and reduces runoff and soil erosion [14]. The potential synergy between these species could operate through three main mechanisms. Spatially, their deep (*P. centrasiaticum*) and shallow (*S. viridis*) root systems can maximize resource acquisition across soil profiles [57,58,59,60,61]. Temporally, rapid annual cover by *S. viridis* can complement the perennial growth of *P. centrasiaticum*. Functionally, *S. viridis* can stabilize topsoil while *P. centrasiaticum* improves the deep soil structure. This strategy is supported by global evidence that diverse plant communities enhance survival and reduce erosion in rehabilitated mine sites [8]. However, since the proposed model has not yet been verified under field conditions, it should be regarded as hypothesis-generating. Further investigation is needed to optimize species ratios and evaluate interspecific competition, in order to balance ecological benefits with long-term community stability.

## 5. Conclusions

This study demonstrates that successful early-stage phytoremediation of arid limestone tailings requires an integrated strategy addressing substrate limitations, water scarcity, and specific nutrient constraints. Our experiment yielded three key findings from which practical implications have emerged. Firstly, the 2:1 soil-to-tailings substrate ratio provided the optimal physical structure for initial plant establishment, while water availability was the overriding limiting factor in this arid environment, directly controlling physiological performance and biomass. Secondly, tailings inherently supplied sufficient nitrogen, rendering N-fertilization ineffective, whereas potassium (K) was identified as a critical, potentially yield-limiting nutrient that requires targeted management. Thirdly, by integrating these factors, we proposed a practical restoration model based on the complementary use of native species—the deep-rooted perennial *P. centrasiaticum* and the shallow-rooted pioneer *S. viridis*. This model was designed to exploit spatial and temporal niche differentiation to enhance resource use and ecosystem stability.

It is important to acknowledge the limitations of this study to properly contextualize its findings. Firstly, the research was conducted as a single-season greenhouse pot experiment with a relatively small sample size. The results may not fully capture the long-term dynamics, species interactions, and environmental stresses encountered in field conditions. Secondly, we utilized tailings and topsoil from a single source. Limestone tailings from different geological formations may vary in terms of geochemical properties, which could influence plant responses and optimal amendment strategies. Thirdly, soil physical analyses and microbial assays were not directly measured. Therefore, the framework provides a science-based strategy for restoring similar arid limestone tailings. The proposed species combination and its implied synergies required validation under field conditions, particularly regarding long-term interspecific interactions and productivity. Furthermore, the model’s applicability to tailings with fundamentally different geochemistry or to more humid climates necessitates separate investigation. Future research should prioritize multi-year field trials, the inclusion of a broader range of tailings, and direct measurement of soil structural and microbial parameters to advance toward robust, scalable restoration protocols.

## Figures and Tables

**Figure 1 biology-15-00082-f001:**
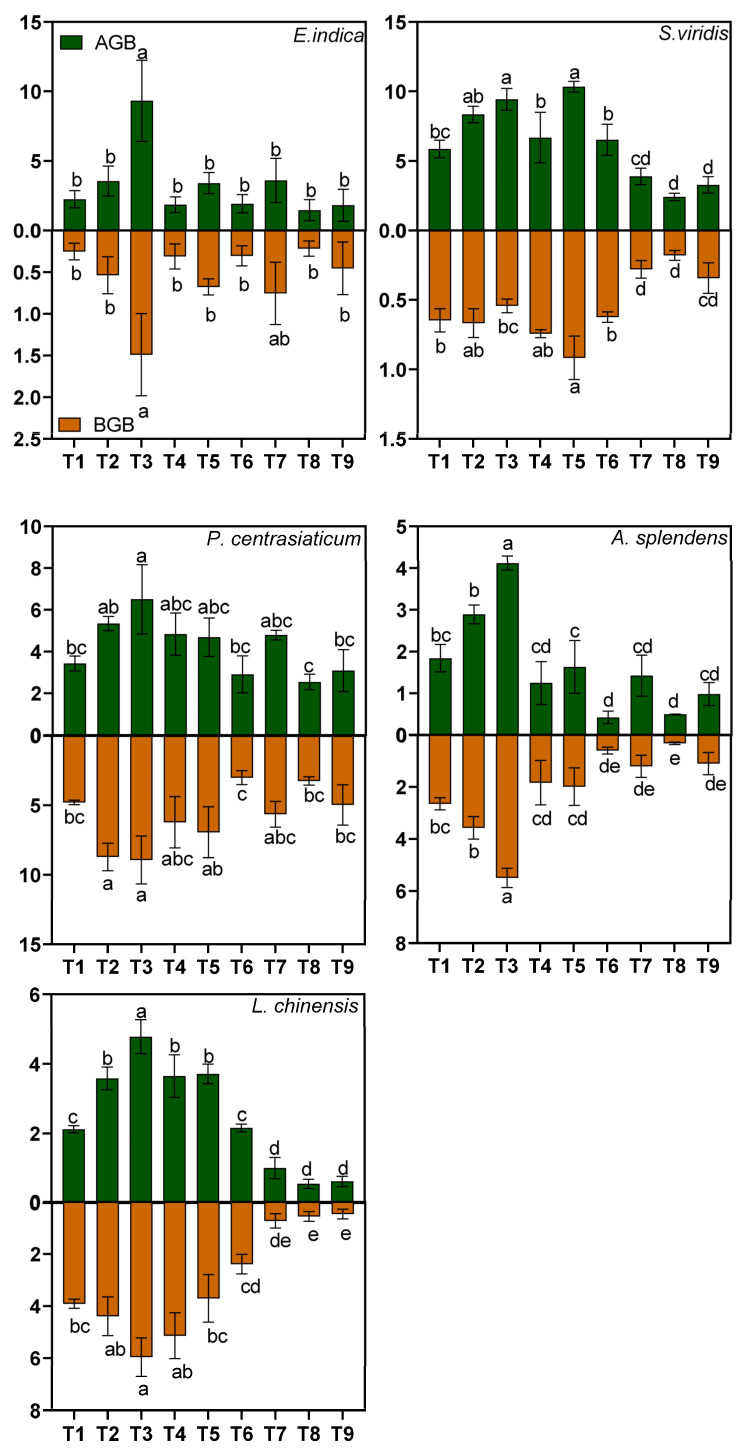
Effects of treatments on the aboveground (AGB), belowground (BGB), and total biomass (TB) of five plant species. Lowercase letters indicate significant differences among treatments for each species (*p* < 0.05, Tukey’s HSD test). Error bars represent ±1 SE (n = 3).

**Figure 2 biology-15-00082-f002:**
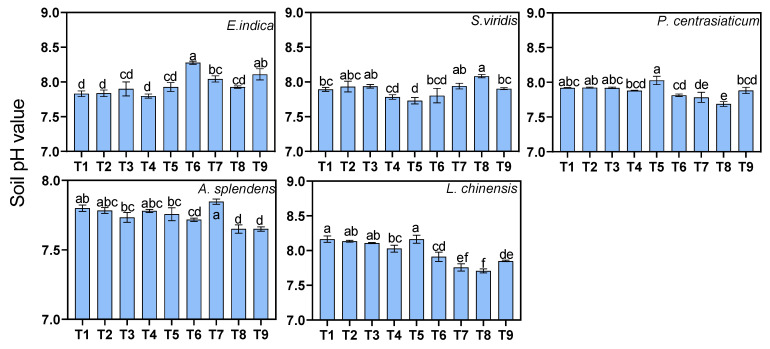
Effects of treatment on soil pH for each plant species. Lowercase letters indicate significant differences among treatments within species (*p* < 0.05, Tukey’s HSD test). Error bars represent ±1 SE (n = 3).

**Figure 3 biology-15-00082-f003:**
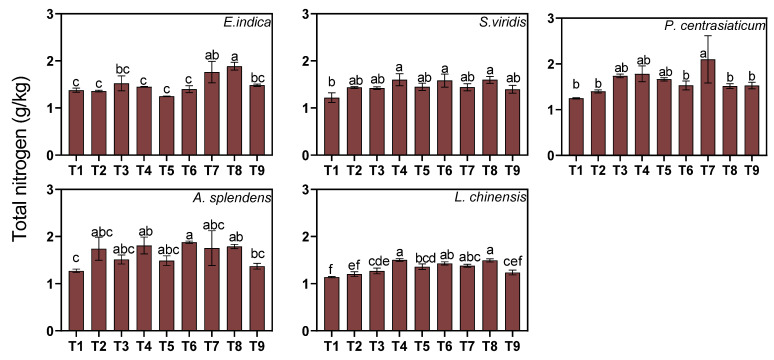
Effects of the treatments on soil total nitrogen (TN). Lowercase letters indicate significant differences among treatments within each nutrient (*p* < 0.05, Tukey’s HSD test). Error bars represent ±1 SE (n = 3).

**Figure 4 biology-15-00082-f004:**
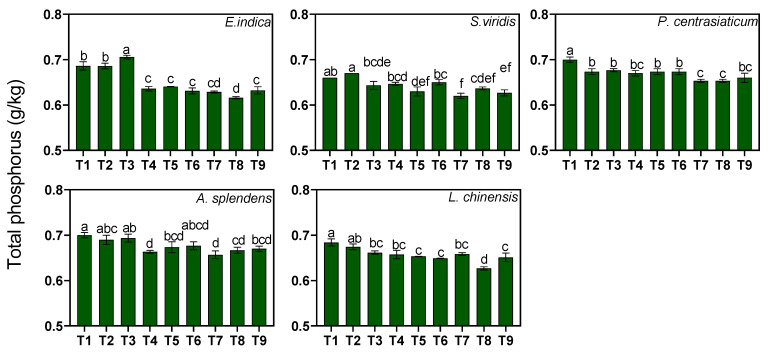
Effects of the treatments on soil total phosphorus (TP) concentrations. Lowercase letters indicate significant differences among treatments within each nutrient (*p* < 0.05, Tukey’s HSD test). Error bars represent ±1 SE (n = 3).

**Figure 5 biology-15-00082-f005:**
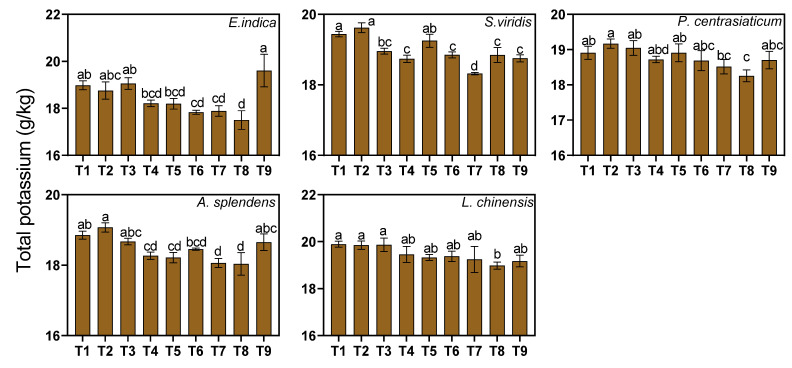
Effects of the treatments on soil total potassium (TK) concentrations. Lowercase letters indicate significant differences among treatments within each nutrient (*p* < 0.05, Tukey’s HSD test). Error bars represent ±1 SE (n = 3).

**Figure 6 biology-15-00082-f006:**
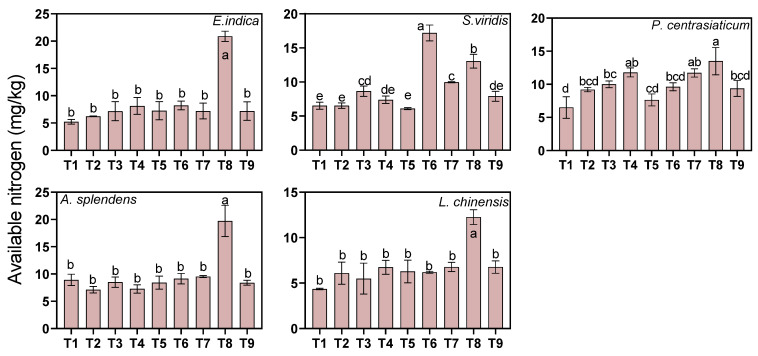
Effects of the treatments on soil available nitrogen (AN). Lowercase letters indicate significant differences among treatments within each nutrient (*p* < 0.05, Tukey’s HSD test). Error bars represent ±1 SE (n = 3).

**Figure 7 biology-15-00082-f007:**
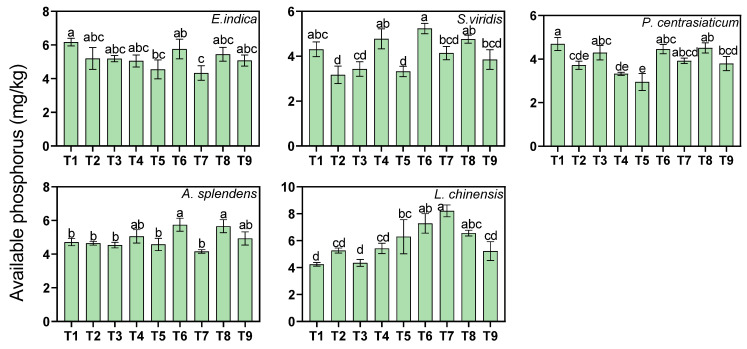
Effects of the treatments on soil available phosphorus (AP) concentrations. Lowercase letters indicate significant differences among treatments within each nutrient (*p* < 0.05, Tukey’s HSD test). Error bars represent ±1 SE (n = 3).

**Figure 8 biology-15-00082-f008:**
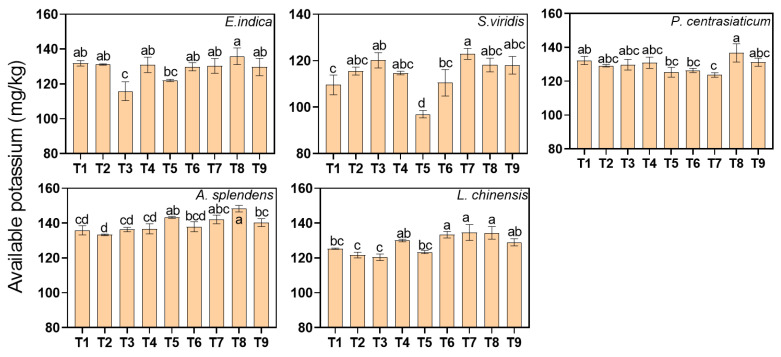
Effects of the treatments on soil available potassium (AK) concentrations. Lowercase letters indicate significant differences among treatments within each nutrient (*p* < 0.05, Tukey’s HSD test). Error bars represent ±1 SE (n = 3).

**Figure 9 biology-15-00082-f009:**
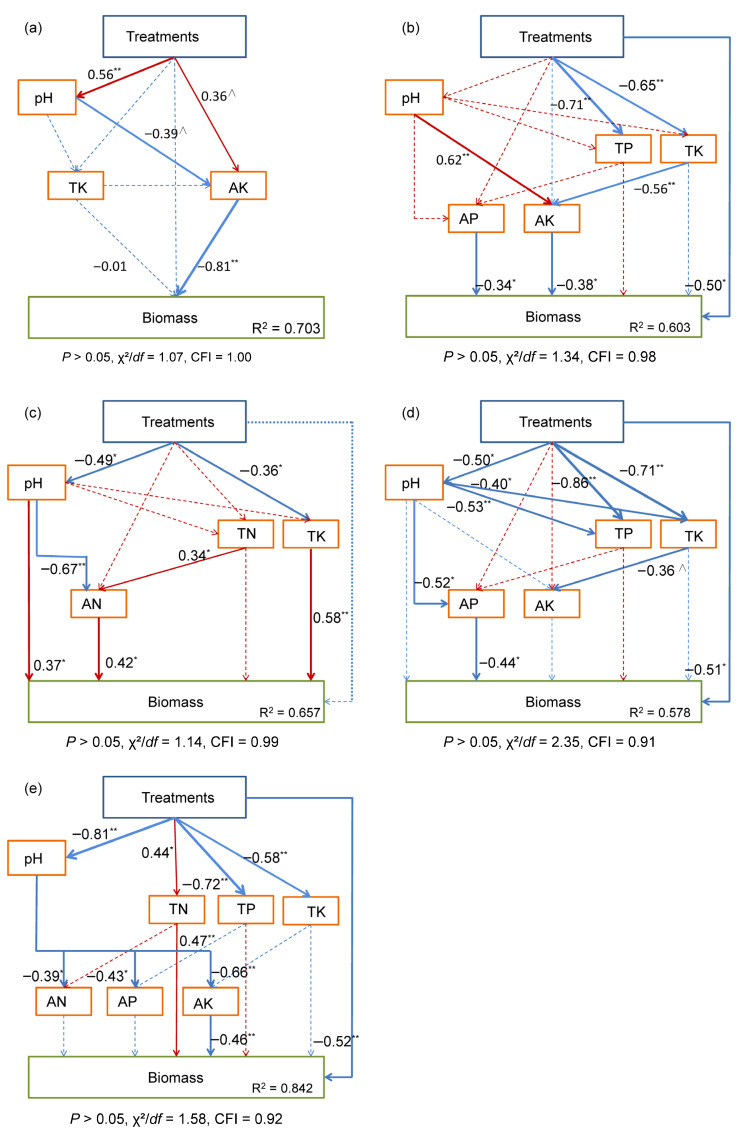
Structural equation model of the effects of the treatments on plant biomass accumulation. (**a**) *E. indica*, (**b**) *S. viridis*, (**c**) *P. centrasiaticum*, (**d**) *A. splendens*, and (**e**) *L. chinensis*. Standardized path coefficients are indicated by solid arrows (*p* < 0.05) and dashed arrows (*p* > 0.05). ** *p* < 0.001, * *p* < 0.05, and ∧ *p* < 0.10.

**Table 1 biology-15-00082-t001:** Orthogonal test table.

Experimental Group	Factors			Configuration Combination
	A: Substrate Composition	B: Soil Moisture, %	C: Nitrogen Fertilization, g N·km^−2^	
T_1_	Soil:tailings = 2:1	30	0	A_1_B_1_C_1_
T_2_	Soil:tailings = 2:1	45	200	A_1_B_2_C_2_
T_3_	Soil:tailings = 2:1	60	400	A_1_B_3_C_3_
T_4_	Soil:tailings = 1:1	45	400	A_2_B_2_C_3_
T_5_	Soil:tailings = 1:1	60	0	A_2_B_3_C1
T_6_	Soil:tailings = 1:1	30	200	A_2_B_1_C_2_
T_7_	Soil:tailings = 1:2	60	200	A_3_B_3_C_2_
T_8_	Soil:tailings = 1:2	30	400	A_3_B_1_C_3_
T_9_	Soil:tailings = 1:2	45	0	A_3_B_2_C_1_

**Table 2 biology-15-00082-t002:** The range analysis of orthogonal experimental results regarding biomass.

	*E. indica*			*S. viridis*			*P. centrasiaticum*			*A. splendens*			*L. chinensis*		
	A	B	C	A	B	C	A	B	C	A	B	C	A	B	C
K_1_	52.23	19.21	26.58	76.43	48.77	64.14	113.18	59.79	83.74	61.68	19.01	30.58	31.47	14.50	19.36
K_2_	25.46	25.60	32.06	77.47	60.16	60.97	85.81	99.53	91.23	23.19	34.89	30.35	28.56	23.56	20.25
K_3_	24.94	57.81	43.99	31.17	76.14	59.96	72.82	112.49	96.84	16.62	47.59	40.56	6.53	28.50	26.95
k_1_	5.80	2.13	2.95	8.49	5.42	7.13	12.58	6.64	9.30	6.85	2.11	3.40	3.50	1.61	2.15
k_2_	2.83	2.84	3.56	8.61	6.68	6.77	9.53	11.06	10.14	2.58	3.88	3.37	3.17	2.62	2.25
k_3_	2.77	6.42	4.89	3.46	8.46	6.66	8.09	12.50	10.76	1.85	5.29	4.51	0.73	3.17	2.99
R	3.03	4.29	1.93	5.03	3.04	0.46	4.48	5.86	1.46	5.01	3.18	1.13	2.77	1.56	0.84
Optimal Level	1	3	3	2	3	1	1	3	3	1	3	3	1	3	3

Note: Factor codes: A (substrate composition), B (soil moisture regime), and C (nitrogen fertilization). K_1_, K_2_, K_3_: Sum of the response values at level 1, 2, and 3 of each factor, respectively. k_1_, k_2_, k_3_: Mean response (*K_i_*/3) at each level. R: Range (max(k_i_) − min(k_i_)), indicating the magnitude of the factor’s effect.

**Table 3 biology-15-00082-t003:** Results of stepwise multiple regression analyzing the effects of soil pH and nutrient characteristics on plant biomass components.

Species	Dependent Variable	Predictor Variable	Standardized β	*p*-Value	Model Fit Adjusted R^2^
	AGB	AK	−0.825	<0.001	0.667
*E. indica*	BGB	AK	−0.813	<0.001	0.647
	TB	AK	−0.823	<0.001	0.679
	AGB	AP	−0.434	<0.05	0.349
		AK	−0.404	<0.05	
*S. viridis*	BGB	PH	−0.615	<0.001	0.534
		TK	0.397	<0.05	
	TB	AK	−0.427	<0.05	0.361
		AP	−0.423	<0.05	
	AGB	TK	0.603	<0.05	0.338
*P. centrasiaticum*	BGB	TK	0.543	<0.05	0.267
	TB	TK	0.589	<0.05	0.320
	AGB	AP	−0.458	<0.05	0.403
		AK	−0.391	<0.05	
*A. splendens*	BGB	AK	−0.568	<0.05	0.295
	TB	AK	−0.455	<0.05	0.386
		AP	−0.379	<0.05	
	AGB	pH	0.757	<0.001	0.557
*L. chinensis*	BGB	pH	0.726	<0.001	0.508
	TB	pH	0.756	<0.001	0.555

**Table 4 biology-15-00082-t004:** Principal component analysis of evaluation indicators for restorative plants.

Variables	PC 1	PC 2	PC 3	PC 4
Aboveground biomass	0.774	0.383	−0.155	−0.331
Belowground biomass	0.567	0.209	0.597	0.422
Biomass	0.853	0.378	0.27	0.048
Soil pH	0.331	0.022	−0.436	0.711
Soil total carbon content	−0.427	0.616	−0.183	0.375
Soil total nitrogen content	−0.36	0.595	0.379	0.055
Soil total phosphorus content	0.344	−0.431	0.694	−0.085
Soil total potassium content	0.635	−0.445	−0.115	0.128
Soil available nitrogen content	−0.502	0.474	0.16	−0.196
Soil available phosphorus content	−0.458	−0.559	−0.08	0.148
Soil available potassium content	−0.611	−0.238	0.602	0.264
Eigenvalue	3.427	2.044	1.710	1.093
Contribution rate	31.155%	18.580%	15.541%	9.936%
Cumulative contribution rate	31.155%	49.735%	65.276%	75.212%

**Table 5 biology-15-00082-t005:** Comprehensive evaluation of different restorative plants.

Species	Comprehensive Score	Ranking
*E. indica*	−1.338055576	5
*S. viridis*	−0.377783044	2
*P. centrasiaticum*	0.432640691	1
*A. splendens*	−1.287534172	4
*L. chinensis*	−0.916406638	3

## Data Availability

The data presented in this study are available on request from the corresponding author.

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
