# Peer review of "Ecological Restoration of Limestone Tailings in Arid Regions: A Synergistic Substrate–Plant Approach"

_biology, 2025, doi:10.3390/biology15010082_

Round 1
Reviewer 1 Report
Comments and Suggestions for Authors
- Limstone tailing sampling method need to modifiy
- Native plant seed sampling proceedure need more eloboration
- elaborate the different water holding capacities procedure more clearly
- need more details on soil properties, which was used to grow seedlings in greenhouse
- Use appropriate methods for soil analysis anlong with references
- Results are well presented
- Discussion need to improve throughly and link with obtained results
- Describe potential limitations of the study
- Conclusion is not well written especially "concluding remarks" need to comprehensively modify.
Need to improve throughout manuscript
Author Response
Reviewer 1
We would like to extend our most sincere gratitude for your thorough review and the highly constructive feedback provided on our manuscript. Your comments on the Materials and Methods section were invaluable, guiding us to significantly enhance the clarity, completeness, and reproducibility of our experimental descriptions. Similarly, your insights on the Discussion section prompted a deeper and more focused analysis, strengthening the link between our results and their broader implications.
We have carefully addressed all points raised in the revision. It is our hope that these modifications have made the manuscript more rigorous, coherent, and compelling. We are also truly encouraged by your positive remarks on certain aspects of our work. Your acknowledgment has given us great confidence and is deeply appreciated.
Thank you once again for your time, expertise, and contribution to improving our paper.
Comments 1: Limstone tailing sampling method need to modifiy
Response 1: Thanks for your advice. “Limestone tailings samples were collected from three 10 m × 10 m plots located in the upstream, midstream, and downstream sections of the accumulation zones at the eastern foot of the Taihang Mountains (38°15′N, 114°32′E; elevation 410 m). Plots were selected based on their undisturbed conditions, considering both vegetation coverage and spatial representativeness. A diagonal sampling method was employed: five 0–20 cm sub-samples (collected from the four corners and the center of each plot) were thoroughly mixed to form a single 2 kg composite sample per plot, thereby reducing heterogeneity. On-site sieving using 0.5 mm and 10 mm stainless steel sieves ensured a particle size range of 0.5–1 cm, and any visible debris was manually removed” We have added this information and highlighted it on Page 4. We sincerely thanks for this suggestion, which has helped us make the methodology in our manuscript more precise and its justification more transparent.
Comments 2: Native plant seed sampling proceedure need more elaboration
Response 2: Thanks for your suggestion. “Seeds of five native plant species were collected near the limestone mine wastelands in 2020. Mature, healthy seeds were selectively harvested from at least 50 randomly chosen individual plants per species to capture genetic diversity. After collection, seeds were air‑dried at room temperature, cleaned to remove debris, and stored in paper bags at 4°C in the dark until the initiation of the experiment in 2023”. We have added this information and highlighted it on Page 4.
Comments 3: elaborate the different water holding capacities procedure more clearly
Response 3: Thanks for your suggestion. Soil moisture levels were maintained at 30%, 45%, and 60% of the soil's maximum water‑holding capacity (WHC). The procedure began on August 15 and was conducted daily throughout the experiment. Each pot was weighed using a digital balance. The amount of deionized water required to restore the pot to its predetermined target weight (corresponding to the specific WHC percentage for that treatment) was then calculated and added. This gravimetric method ensured precise and consistent moisture control across all replicates. We have added this information and highlighted it on Page 5.
Comments 4: need more details on soil properties, which was used to grow seedlings in greenhouse
Response 4: Thanks for your suggestion. The seedlings were raised in a greenhouse. The potting matrix used for this nursery stage consisted of pure, unamended soil, which was identical to the soil component described in Section 2.1 and contained no tailings or other additives. We have added this information and highlighted it on Page 5. Thank you for your suggestions regarding the experimental methodology. They have made our research more rigorous, detailed, and reproducible.
Comments 5: Use appropriate methods for soil analysis anlong with references
Response 5: Thanks for this suggestion to ensure methodological rigor. We have thoroughly revised the “Soil Physicochemical Analysis” subsection (Section 2.5) to explicitly state the standard methods used for each measurement, accompanied by their corresponding authoritative references. The key additions are as follows:
Soil pH was measured in a 1:2.5 (w/v) soil-water suspension. Total nitrogen (TN) was quantified using the Kjeldahl digestion method (Bremner, 1996). Available phosphorus (AP) was extracted with 0.5 M NaHCO₃ (pH 8.5) and measured by molybdenum blue colorimetry (Olsen et al., 1954). Total potassium (TK) was quantified by flame photometry after digestion with HF–HClO₄. Available potassium (AK) was extracted with 1 M NH₄OAc (pH 7.0) and measured by flame photometry (Knudsen et al., 1982). Alkali-hydrolyzable nitrogen (AN) was determined by the alkaline diffusion method (Bao, 2000).
All analyses were performed on the composite soil samples (one per pot, n=27). These revisions provide a complete and citable account of our analytical procedures, ensuring reproducibility and aligning with established soil science practices. We have revised and highlighted them on Page 5-6. Thanks again for your suggestion.
Comments 6: Results are well presented
Response 6: Thanks for your affirmation.
Comments 7: Discussion need to improve throughly and link with obtained results
Comments 8:Describe potential limitations of the study
Comments 9: Conclusion is not well written especially "concluding remarks" need to comprehensively modify.
Response 7-9: Thank you for these constructive comments. We have thoroughly revised the manuscript in direct response to your points.
The Discussion section has been comprehensively rewritten to provide a deeper and more logical interpretation of our findings. A primary focus was restructuring the analysis of soil potassium (K) dynamics (Section 4.2, now titled "Shifts in Potassium Pool Equilibrium Driven by Plant-Soil Interactions"). We now clearly link the observed correlations (TK-biomass positive, AK-biomass negative) to plant-soil feedbacks under different growth conditions, rather than overstating K as a primary limiting factor.
As suggested, we have added a paragraph in Conclusion to present the limitations and future perspectives. This section explicitly addresses key limitations, including the use of a single-season pot experiment, the single source of tailings and soil, the lack of direct microbial or soil physical data, and the sample size constraints for certain analyses. The entire Conclusion section has been rewritten for greater clarity and precision. We have removed overgeneralized statements, strictly limited the scope of our findings to the experimental conditions, reframed the proposed restoration model as a hypothesis for future field validation, and integrated the discussion on limitations as noted above. We believe these revisions have significantly strengthened the manuscript's rigor, clarity, and scholarly contribution. Thank you again for your invaluable guidance.
Reviewer 2 Report
Comments and Suggestions for Authors
This study provides a systematic evaluation of the effects of substrate amendments and native plant species on the ecological restoration of limestone tailings in arid northern China. The research objectives are clearly articulated, the experimental design is generally robust, and the analytical methods are appropriate, giving the work both theoretical value and practical relevance. Nevertheless, the manuscript requires further refinement regarding figure and table presentation, methodological completeness, and linguistic quality to enhance its scientific rigor and readability.
Major Suggestions:
- The figures in their current form appear insufficiently polished. It is recommended to improve color schemes, axis labels, and legend clarity to meet academic publishing standards. More distinguishable graphical symbols should be adopted, and figures should remain interpretable in grayscale to ensure accessibility across print formats.
- Essential geographic information about the experimental site (e.g., coordinates, climate classification, soil characteristics) is missing and should be added. Including a schematic diagram of the experimental layout would effectively illustrate the design and spatial distribution of treatment groups, thereby improving reader understanding.
- The Results section could offer clearer interpretation of statistical outcomes to better emphasize major findings. The Discussion section would benefit from deeper engagement with relevant literature to further explore underlying mechanisms and to better highlight the study’s innovation and limitations.
Several sentences are lengthy, repetitive, or ambiguous. A comprehensive language polish is recommended to enhance conciseness and precision. Consistency in the use of technical terminology should be carefully reviewed, and informal expressions should be avoided throughout the manuscript.
Author Response
Comments :This study provides a systematic evaluation of the effects of substrate amendments and native plant species on the ecological restoration of limestone tailings in arid northern China. The research objectives are clearly articulated, the experimental design is generally robust, and the analytical methods are appropriate, giving the work both theoretical value and practical relevance. Nevertheless, the manuscript requires further refinement regarding figure and table presentation, methodological completeness, and linguistic quality to enhance its scientific rigor and readability.
Response: We would like to extend our most sincere gratitude for your thorough review and the highly constructive feedback provided on our manuscript. We are also truly encouraged by your positive remarks on certain aspects of our work. Your acknowledgment has given us great confidence and is deeply appreciated. Your comments were important, guiding us to significantly enhance the clarity, completeness, and reproducibility of our experimental descriptions, and to prompt a deeper and more focused analysis, strengthening the link between our results and their broader implications.
Thank you once again for your time, expertise, and contribution to improving our paper.
Major Suggestions:
Comments 1: The figures in their current form appear insufficiently polished. It is recommended to improve color schemes, axis labels, and legend clarity to meet academic publishing standards. More distinguishable graphical symbols should be adopted, and figures should remain interpretable in grayscale to ensure accessibility across print formats.
Response 1: Thanks for your comment. In accordance with your recommendations, we have comprehensively revised all figures. The improvements specifically address the color schemes, axis labels, and legend clarity to meet academic publishing standards. We have also adopted more distinguishable graphical symbols and ensured that all figures remain fully interpretable when printed in grayscale.
Comments 2: Essential geographic information about the experimental site (e.g., coordinates, climate classification, soil characteristics) is missing and should be added. Including a schematic diagram of the experimental layout would effectively illustrate the design and spatial distribution of treatment groups, thereby improving reader understanding.
Response 2: Thanks for these constructive suggestions to improve the clarity of our experimental setup.
“Limestone tailings samples were collected from three 10 m × 10 m plots located in the upstream, midstream, and downstream sections of the accumulation zones at the eastern foot of the Taihang Mountains (38°15′N, 114°32′E; elevation 410 m). Plots were selected based on their undisturbed conditions, considering both vegetation coverage and spatial representativeness. A diagonal sampling method was employed: five 0–20 cm sub-samples (collected from the four corners and the center of each plot) were thoroughly mixed to form a single 2 kg composite sample per plot, thereby reducing heterogeneity. On-site sieving using 0.5 mm and 10 mm stainless steel sieves ensured a particle size range of 0.5–1 cm, and any visible debris was manually removed.
Topsoil samples were collected from three 10 m × 10 m control plots, situated ap-proximately 500 m from the mine boundary, where soil and topography were consistent with the regional background (the eastern foot of the Taihang Mountains—38°14′N, 114°31′E; elevation 390 m). The same diagonal sampling method was applied (five 0–20 cm sub-samples were mixed into one 2 kg composite sample per plot). All tailings and topsoil samples were transported to the laboratory within 24 hours under cooled conditions (4°C) and stored at 4°C following air-drying and homogenization. The field-collected topsoil used as the planting medium had the following mean properties: pH—7.30; organic matter content—21.30 g/kg; total nitrogen content—1.92 g/kg; total phosphorus—0.86 content g/kg; total potassium content—25.03 g/kg; alkali-hydrolyzable nitrogen content—51.47 mg/kg; available phosphorus content—4.32 mg/kg; and available potassium content—148.45 mg/kg”.
We have added essential geographic information about the experimental site, soil characteristics, and a detailed description of the sampling methodology in the revised version. Once again, we are grateful for your advice, which has made our paper more comprehensive, rigorous, and reproducible.
Comments 3: The Results section could offer clearer interpretation of statistical outcomes to better emphasize major findings. The Discussion section would benefit from deeper engagement with relevant literature to further explore underlying mechanisms and to better highlight the study’s innovation and limitations.
Response 3: Thank you for this valuable suggestion. Following your advice, we have thoroughly rewritten the Results section to provide a clearer interpretation of the statistical outcomes, ensuring that the major findings are emphasized. These revisions have been highlighted in the manuscript. The Discussion section has also been revised accordingly. The primary focus of this revision is on the interpretation of soil potassium dynamics. Furthermore, a dedicated discussion of the study's limitations has been incorporated into the Conclusion section. We have highlighted them in revised manuscript. We are sincerely grateful for your guidance, which has significantly enhanced the clarity of our results and the depth of our discussion.
Comments on the Quality of English Language
Comments 4: Several sentences are lengthy, repetitive, or ambiguous. A comprehensive language polish is recommended to enhance conciseness and precision. Consistency in the use of technical terminology should be carefully reviewed, and informal expressions should be avoided throughout the manuscript.
Response 4: Thank you for your reminder. We have completed professional language editing through MDPI Author Services, rewritten the relevant paragraphs accordingly, and corrected these errors. Once again, thanks for your reminder, which has made our paper more rigorous and readable.
Reviewer 3 Report
Comments and Suggestions for Authors
Restoring vegetation on limestone tailings in arid and semi-arid regions remains a major ecological challenge due to severe nutrient limitations, poor substrate structure, and the low water-holding capacity that restrict plant establishment and long-term ecosystem recovery.
Abstract
- The Simple Summary and Abstract provide a general overview, but lack essential quantitative details.
- The description of the experimental design is somewhat scattered. The orthogonal L9 design, three factors (substrate ratio, moisture level, N addition), five species, and the measured variables should be presented more systematically.
- Some statements are overgeneralized. Claims such as providing a “science-based framework for restoring arid mining ecosystems” should be limited to the specific conditions studied (limestone tailings, controlled pot experiment, northern China).
- Keywords are too broad and do not reflect the paper’s ecological context. More specific terms (e.g., limestone tailings, arid regions, native grasses, Pennisetum centrasiaticum) should be added.
Introduction
- The Introduction lacks site-specific background. No information is given on the climate, precipitation, temperature regime, geological context, or soil characteristics of the limestone tailings area—details important for evaluating ecological relevance and transferability.
- Several conceptual points (e.g., limitations of soil transfer, benefits of in situ reconstruction) are repeated in multiple paragraphs, making the narrative overly long. The text could be more concise.
- The study objectives are listed, but no clear hypotheses are provided. Hypotheses such as (i) optimal biomass under 2:1 tailings:soil + 60% moisture, (ii) K as a potential limiting nutrient, and (iii) superior performance of C4 species such as P. centrasiaticum should be stated explicitly.
Methods
- Site description is missing: geographic coordinates, elevation, long-term rainfall, temperature, soil type, and baseline tailings properties (pH, TN, TP, TK, texture) should be included.
- Essential experimental details are incomplete: pot size/volume, number of seedlings per pot, spacing, greenhouse temperature and humidity, photoperiod, and the total duration of the experiment. Without this, reproducibility is limited.
- Sampling details are unclear: number of subsamples, number of technical replicates, and whether composite samples were used.
- The statistical analysis section lacks justification for using three-way ANOVA with an L₉ orthogonal array (which is not a full factorial design). Model assumptions and the implications of unbalanced treatment combinations should be addressed.
- Stepwise regression and SEM descriptions are incomplete: the criteria for variable selection, checks for multicollinearity, sample size adequacy, and additional SEM fit indices (e.g., RMSEA, SRMR) should be reported.
Results
- Biomass and nutrient results include excessive numerical detail. Patterns should be emphasized, with only essential percentages retained in the text; the rest is already in figures and tables.
- Figure readability is limited: axis labels, units, and font sizes are small. Graphs with five species and nine treatments are visually dense and need simplification.
- Table 2 contains abbreviations (K1, K2, k1, R, etc.) that should be clearly defined within the table caption, not only in the text.
- SEM diagrams include too many arrows and parameters relative to sample size. A simplified model focusing on major pathways would improve clarity.
- PCA and comprehensive scoring lack clear explanation of normalization and weighting procedures; a supplemental formula or workflow would improve transparency.
Discussion
- The discussion occasionally overgeneralizes results from a single-season greenhouse pot experiment to “arid mining ecosystems” at large. Conclusions should be limited to limestone tailings and similar controlled conditions.
- Limitations of the study are not discussed. Important limitations include: greenhouse pot conditions rather than field conditions, short experimental duration, single source of tailings and topsoil, lack of microbial and soil-structural data, SEM conducted with relatively small sample size.
- The “complementary model” proposed for P. centrasiaticum and S. viridis is reasonable, but should be clearly stated as hypothesis-generating, since it was not verified under field conditions.
- The Conclusions section repeats parts of the Abstract almost verbatim; it should instead highlight only 3–4 core messages (optimal substrate/moisture combination, key role of K, species selection, and practical restoration implications).
Author Response
Comments: Restoring vegetation on limestone tailings in arid and semi-arid regions remains a major ecological challenge due to severe nutrient limitations, poor substrate structure, and the low water-holding capacity that restrict plant establishment and long-term ecosystem recovery.
Response: We would like to express our deepest gratitude for your comprehensive and insightful review. Your constructive comments on the description of the experimental design, the details of the statistical analyses, the integration and presentation of results, and the discussion of the study's limitations have been invaluable. Your suggestions have not only made our manuscript significantly more rigorous, clear, and well-structured but have also provided essential guidance that will positively influence the design and execution of our future research work. We sincerely appreciate the time and expertise you dedicated to improving our paper.
- Abstract
Comments 1.1: The Simple Summary and Abstract provide a general overview, but lack essential quantitative details.
Response 1.1: Thank you for pointing this out. As suggested, we have integrated key quantitative details from the experimental design directly into the Simple Summary and Abstract to make it more informative. We have highlighted them on Page 1. This enhancement allows the Simple Summary and Abstract to better highlight the main findings of our paper. We are grateful for your guidance.
Comments 1.2: The description of the experimental design is somewhat scattered. The orthogonal L9 design, three factors (substrate ratio, moisture level, N addition), five species, and the measured variables should be presented more systematically.
Response 1.2: We appreciate your suggestion for improving clarity. We have reorganized the description of the experimental design into a single, consolidated sentence for better readability and systematic presentation. “An orthogonal L9(3⁴) experimental design was employed to test three factors: the soil-to-tailings ratio (1:2, 1:1, and 2:1), moisture level (30%, 45%, and 60% of field capacity), and nitrogen addition (0, 5, and 10 g N m⁻²). Five native grass species (Pennisetum centrasiaticum, Setaria viridis, Leymus chinensis, Achnatherum splendens, and Eleusine indica) were grown under these treatment conditions, and plant biomass and key soil nutrient variables were measured. Stepwise regression, structural equation modeling, and principal component analysis were applied to assess plant growth responses and soil nutrient dynamics”. We have revised and highlighted it in Abstract on Page 1. We extend our sincere thanks once again for your suggestions, which have significantly enhanced the readability of our paper.
Comments 1.3: Some statements are overgeneralized. Claims such as providing a “science-based framework for restoring arid mining ecosystems” should be limited to the specific conditions studied (limestone tailings, controlled pot experiment, northern China).
Response 1.3: Thanks for this important comment. We agree that our conclusions should be carefully constrained to the scope of the study. We have revised the text throughout to clearly specify the experimental conditions. Specifically, we added “using a controlled pot experiment” in the second sentence to clarify the study setup. And we modified the concluding statement from a broad claim of providing “a science-based framework for rehabilitating mining ecosystems in arid regions” to a more precise description stating that this work provides “fundamental data and a conceptual framework for rehabilitating arid limestone tailings in similar ecological settings, based on controlled experimental evidence.” Furthermore, we added the qualifier “under these experimental conditions” when summarizing the key factors for successful restoration. We have revised them and highlighted in Abstract. These changes would ensure the conclusions are accurately presented as derived from and applicable to the specific context of the study. We sincerely appreciate your rigorous review, which has significantly enhanced the practical relevance of our research.
Comments 1.4: Keywords are too broad and do not reflect the paper’s ecological context. More specific terms (e.g., limestone tailings, arid regions, native grasses, Pennisetum centrasiaticum) should be added.
Response 1.4: Thanks for your suggestion. Following your advice, we have revised the keywords. We acknowledge that our original keywords did not accurately reflect the ecological research context of the paper. The updated set now properly encapsulates the background of our study, which we believe will significantly enhance other readers' understanding of our work. We sincerely appreciate your guidance.
- Introduction
Comments 2.1: The Introduction lacks site-specific background. No information is given on the climate, precipitation, temperature regime, geological context, or soil characteristics of the limestone tailings area—details important for evaluating ecological relevance and transferability.
Response 2.1: Thanks for your suggestion. We have added site-specific background in Introduction. “To address these interrelated gaps, we employed a controlled pot experiment using limestone tailings from an arid region in the eastern foothills of the Taihang Mountains in Northern China (38°15′N, 114°32′E). This area is characterized by a warm-temperate, semi-humid to semi-arid continental monsoon climate, with an average annual temperature of 8–14°C and an annual precipitation of approximately 500 mm, concentrated predominantly in the summer months. The zonal soil type was limestone soil and some was shale soil.” We have added and highlighted it on Page 3. We sincerely thank you once again for this suggestion. Your reminder has been instrumental in making our manuscript more rigorous, contextualized, and valuable for readers seeking to understand the transferability of restoration strategies to similar arid, calcareous environments.
Comments 2.2: Several conceptual points (e.g., limitations of soil transfer, benefits of in situ reconstruction) are repeated in multiple paragraphs, making the narrative overly long. The text could be more concise.
Response 2.2: We have significantly condensed the introduction, removing repetitive statements about general principles. While the core focus of the introduction is to establish the conceptual gap regarding limestone tailings, we have added a sentence in the final paragraph specifying that the experiment uses tailings "from an arid region in northern China," providing necessary context. Thanks again for your advice. It makes our manuscript more concise.
Comments 2.3: The study objectives are listed, but no clear hypotheses are provided. Hypotheses such as (i) optimal biomass under 2:1 tailings:soil + 60% moisture, (ii) K as a potential limiting nutrient, and (iii) superior performance of C4 species such as P. centrasiaticum should be stated explicitly.
Response 2.3: Thanks for this constructive suggestion. We have revised the Introduction. “Based on this context, we hypothesized that: (i) plant biomass would be maximized under an optimal combination of tailings/soil ratio, water level, and nitrogen level; (ii) potassium (K) availability would be a primary nutrient limitation in this high‑calcium system; and (iii) species with contrasting root architectures and resource‑use strategies (e.g., deep‑rooted perennial vs. shallow‑rooted annual grasses) would exhibit differential performances.” Thank you for your suggestion. It makes the logic of our manuscript much clearer.
- Methods
Comments 3.1: Site description is missing: geographic coordinates, elevation, long-term rainfall, temperature, soil type, and baseline tailings properties (pH, TN, TP, TK, texture) should be included.
Response 3.1: Thanks for your suggestion. We have added Site description in Introduction and Materials and Methods. “The limestone tailings from an arid region in eastern foothills of the Taihang Mountains in northern China (38°15′N, 114°32′E). This area is characterized by a warm-temperate, semi-humid to semi-arid continental monsoon climate, with an average annual temperature of 8–14°C and an annual precipitation of approximately 500 mm, concentrated predominantly in the summer months. The zonal soil type was limestone soil and some was shale soil.” “The field-collected topsoil used as the planting medium had the following mean properties: pH—7.30; organic matter content—21.30 g/kg; total nitrogen content—1.92 g/kg; total phosphorus—0.86 content g/kg; total potassium content—25.03 g/kg; alkali-hydrolyzable nitrogen content—51.47 mg/kg; available phosphorus content—4.32 mg/kg; and available potassium content—148.45 mg/kg”. We have added the information and highlighted them on Page 3 and Page 4. Once again, we are grateful for your advice, which has made our paper more comprehensive, rigorous, and reproducible.
Comments 3.2: Essential experimental details are incomplete: pot size/volume, number of seedlings per pot, spacing, greenhouse temperature and humidity, photoperiod, and the total duration of the experiment. Without this, reproducibility is limited.
Response 3.2: Thank you for highlighting the need for complete experimental details to ensure reproducibility. We have now supplemented the Materials and Methods section with the following information:
Pot specifications: Cylindrical, ~6 L volume (upper diameter: 26.5 cm, lower diameter: 15.5 cm, height: 17.5 cm), with one seedling per pot.
Planting layout: A completely randomized design was used for the 27 pots.
Greenhouse conditions: The experiment was conducted in a rain-sheltered, open-sided greenhouse (i.e., without climate control for temperature and humidity). This setup was intentionally chosen to simulate near-natural field conditions of the local environment while excluding uncontrolled rainfall, thereby ensuring that the manipulated soil moisture treatments were the sole source of variation in water availability. Therefore, the temperature, humidity, and photoperiod inside the greenhouse closely mirrored the ambient conditions of the local study site during the experimental period (May to October 2023).
Experiment duration: The experiment was conducted from seedling transplantation in June 2023 to final harvest in October 2023.
We have added the information and highlighted them in “2.2. The Design of the Experiment” and “2.3. Greenhouse Conditions and Growth Management” on Page 4 and Page 5. We sincerely appreciate your suggestion, which has helped us provide a much clearer and more complete account of our experimental setup.
Comments 3.3: Sampling details are unclear: number of subsamples, number of technical replicates, and whether composite samples were used.
Response 3.3: We apologize for the lack of clarity. We have revised the sampling description in Section 2.4.
“Plants were harvested starting on 15 October. For each of the 27 experimental pots (biological replicates), the following procedure was conducted: Plant height was recorded. The entire aboveground biomass and the complete root system were carefully separated. These constituted one plant sample per pot. Concurrently, soil was collected from the root zone of the same pot. From each pot, three spatially distributed soil subsamples were taken and thoroughly homogenized to form a single composite soil sample per pot. Thus, each pot yielded one composite soil sample and one plant sample for subsequent analysis”.
We have added and highlighted it on Page 5. Thanks for your suggestion, which has enhanced the clarity of our experimental setup description.
Comments 3.4: The statistical analysis section lacks justification for using three-way ANOVA with an L₉ orthogonal array (which is not a full factorial design). Model assumptions and the implications of unbalanced treatment combinations should be addressed.
Response 3.4: We appreciate this important methodological point. We have significantly expanded the statistical analysis section. We explicitly acknowledge that the L₉ design is not a full factorial and does not allow independent estimation of all interactions. We justify the use of ANOVA by stating that our model is appropriately specified to test main effects only, with the unestimated interactions pooled into the residual error term—a valid approach for screening experiments with orthogonal arrays. We note that data were checked for normality and homogeneity of variances to satisfy model assumptions. We added the information and highlighted it in the Statistical analyses Section (Page 6). Thanks for your suggestion, which has enhanced the clarity of our statistical analyses and, consequently, made our results more robust and convincing.
Comments 3.5: Stepwise regression and SEM descriptions are incomplete: the criteria for variable selection, checks for multicollinearity, sample size adequacy, and additional SEM fit indices (e.g., RMSEA, SRMR) should be reported.
Response 3.5: Thanks for requesting these essential details. We have comprehensively revised it. Stepwise regression criteria: Entry (α = 0.05) and removal (α = 0.10) thresholds. Multicollinearity check: Assessment via Variance Inflation Factors (VIF < 10). SEM fit indices: Now reporting a full suite: P, χ²/df, CFI, along with their acceptable thresholds. Sample size note: While not explicitly calculating power, we note that our sample size (n=27) is above the commonly recommended minimum for SEM when using robust estimation methods, and model complexity was kept parsimonious. We added the information and highlighted it in the Statistical analyses Section. Thanks for your suggestion, which has enhanced the clarity of our statistical analyses and, consequently, made our results more robust and convincing.
- Results
Comments 4.1: Biomass and nutrient results include excessive numerical detail. Patterns should be emphasized, with only essential percentages retained in the text; the rest is already in figures and tables.
Response 4.1: Thanks for the critical suggestions to improve data presentation. We have thoroughly revised the Results (3.1).
“All species exhibited a significant reduction in biomass with increasing tailings proportion (Figure 1). Species sensitivity varied substantially, forming a clear tolerance gradient. P. centrasiaticum proved to be the most tolerant, with total biomass (TB) under high tailings reduced by only 35.66% relative to the low-tailings control. In contrast, L. chinensis was the most severely affected, with TB declining by 84.14%. A. splendens also showed high sensitivity (TB reduced by 73.05%), while S. viridis and E. indica displayed intermediate reductions (TB reduced by 59.22% and 52.23%, respectively).
Conversely, increased soil moisture significantly enhanced biomass accumulation across all species (Figure 1). E. indica showed the most dramatic positive response, with TB increasing by 200.75% under high moisture conditions. A. splendens and P. centrasiaticum also exhibited strong, moisture-dependent growth, with TB increasing by 150.35% and 88.15%, respectively. S. viridis and L. chinensis benefited to a more moderate extent (TB increased by 56.12% and 70.21%, respectively).”
We believe these changes make the narrative clearer, direct attention to the biological patterns, and provide full transparency for the statistical results. Thanks again.
Comments 4.2: Figure readability is limited: axis labels, units, and font sizes are small. Graphs with five species and nine treatments are visually dense and need simplification.
Response 4.2: Thanks for your comment. In accordance with your recommendations, we have comprehensively revised all figures. The improvements specifically address the color schemes, axis labels, and legend clarity to meet academic publishing standards. We have also adopted more distinguishable graphical symbols and ensured that all figures remain fully interpretable when printed in grayscale.
Comments 4.3: Table 2 contains abbreviations (K1, K2, k1, R, etc.) that should be clearly defined within the table caption, not only in the text.
Response 4.3: Thanks for your suggestion. K₁, K₂, K₃: Sum of the response values at level 1, 2, and 3 of each factor, respectively. k₁, k₂, k₃: Mean response (Kᵢ/3) at each level. R: Range (max(kᵢ) – min(kᵢ)), indicating the magnitude of the factor's effect. We added the information and highlighted on Page 8.
Comments 4.4: SEM diagrams include too many arrows and parameters relative to sample size. A simplified model focusing on major pathways would improve clarity.
Response 4.4: We have revised the structural equation model (SEM) diagrams in response to your suggestion. The updated figures now employ a simplified visual design: pathways that were statistically non-significant have been visually de-emphasized (e.g., presented with dashed or lighter-weight arrows), and path coefficients have been removed from the diagram itself to reduce visual clutter. This adjustment directly addresses the concern about complexity relative to sample size, resulting in figures that focus reader attention on the major and significant causal pathways, thereby greatly enhancing the clarity and interpretability of the models. We sincerely appreciate your guidance, which has been instrumental in improving the graphical communication of our analytical results.
Comments 4.5: PCA and comprehensive scoring lack clear explanation of normalization and weighting procedures; a supplemental formula or workflow would improve transparency.
Response 4.5: Thanks for your suggestion. We have revised the section. “Principal component analysis (PCA) was performed on a correlation matrix derived from standardized plant growth and soil response variables (mean = 0, standard deviation = 1) to eliminate scale differences. The analysis yielded four principal components (PCs) with eigenvalues > 1, explaining 31.16%, 18.58%, 15.54%, and 9.94% of the total variance, respectively, with a cumulative variance of 75.21% (Table 4).
To integrate the multidimensional information into a single metric of overall performance, a comprehensive evaluation score (F) was calculated for each species using the following weighted summation formula:
where Fi is the comprehensive score for the i-th species, PCij is the score of the i-th species on the j-th principal component, and wj is the weight of the j-th component, defined as its variance contribution rate (i.e., wj = Eigenvaluej / Total Variance).
Thus, the calculation for each species was: F = (PC1 score × 0.3116) + (PC2 score × 0.1858) + (PC3 score × 0.1554) + (PC4 score × 0.0994). The species were then ranked in descending order of their F scores to indicate decreasing overall restoration efficacy (Table 5): P. centrasiaticum (highest score), followed by S. viridis, L. chinensis, A. splendens, and E. indica.”
We have added and highlighted it on Page 17. Thanks for your suggestion, which made our results more robust and convincing.
- Discussion
Comments 5.1: The discussion occasionally overgeneralizes results from a single-season greenhouse pot experiment to “arid mining ecosystems” at large. Conclusions should be limited to limestone tailings and similar controlled conditions.
Response 5.1: Thanks for this critical observation. We agree that our original discussion and conclusion were phrased too broadly. In the revised manuscript, we have carefully qualified our statements throughout the Discussion and Conclusion sections to explicitly limit the scope of our findings. Key phrases such as "under controlled pot conditions," "in this system," and "for the tested species under these conditions" have been added to contextualize the results. Most importantly, we have incorporated a dedicated subsection in the Conclusion to present limitations and future perspectives. We have also reframed the concluding statement to present our work as providing "a science‑based for restoring similar arid limestone tailings under comparable conditions," rather than a general framework for arid mining ecosystems. This revision ensures our conclusions are appropriately constrained to the specific context of our study.
Comments 5.2: Limitations of the study are not discussed. Important limitations include: greenhouse pot conditions rather than field conditions, short experimental duration, single source of tailings and topsoil, lack of microbial and soil-structural data, SEM conducted with relatively small sample size.
Response 5.2: We are grateful to you for highlighting these essential limitations. We have now added a dedicated subsection present limitations and future perspectives in the Conclusion.
“It is important to acknowledge the limitations of this study to properly contextualize its findings. Firstly, the research was conducted as a single‑season greenhouse pot experiment with a relatively small sample size. The results may not fully capture the long‑term dynamics, species interactions, and environmental stresses encountered in field conditions. Secondly, we utilized tailings and topsoil from a single source. Limestone tailings from different geological formations may vary in terms of geochemical properties, which could influence plant responses and optimal amendment strategies. Thirdly, soil physical analyses and microbial assays were not directly measured. Therefore, the framework provides a science-based strategy for restoring similar arid limestone tailings. The proposed species combination and its implied synergies required validation under field conditions, particularly regarding long-term interspecific interactions and productivity. Furthermore, the model’s applicability to tailings with fundamentally different geochemistry or to more humid climates necessitates separate investigation. Future research should prioritize multi‑year field trials, the inclusion of a broader range of tailings, and direct measurement of soil structural and microbial parameters to advance toward robust, scalable restoration protocols”.
We added the information and highlighted it on Page 21. We believe this new section significantly strengthens the manuscript by providing an honest assessment of the study's boundaries, enhancing its credibility, and clearly outlining directions for necessary future research. Thanks for your suggestion again.
Comments 5.3: The “complementary model” proposed for P. centrasiaticum and S. viridis is reasonable, but should be clearly stated as hypothesis-generating, since it was not verified under field conditions.
Response 5.3: Thanks for your reminder. In accordance with your suggestion, we have revised the discussion section on Page 20. “Based on these findings, we propose a restoration model combining P. centrasiaticum and S. viridis. This approach aligns with earlier work showing that multi-species planting in tailings rehabilitation enhance plant survival and reduces runoff and soil erosion. The potential synergy between these species could operate through three main mechanisms. Spatially, their deep (P. centrasiaticum) and shallow (S. viridis) root systems can maximize resource acquisition across soil profiles. Temporally, rapid annual cover by S. viridis can complement the perennial growth of P. centrasiaticum. Functionally, S. viridis can stabilize topsoil while P. centrasiaticum improves the deep soil structure. This strategy is supported by global evidence that diverse plant communities enhance survival and reduce erosion in rehabilitated mine sites. However, since the proposed model has not yet been verified under field conditions, it should be regarded as hypothesis-generating. Further investigation is needed to optimize species ratios and evaluate interspecific competition, in order to balance ecological benefits with long-term community stability”. We added the information and highlighted it on Page 20-21. Your suggestion has made our discussion more rigorous and scientific.
Comments 5.4: The Conclusions section repeats parts of the Abstract almost verbatim; it should instead highlight only 3–4 core messages (optimal substrate/moisture combination, key role of K, species selection, and practical restoration implications).
Response 5.4: Thanks for this constructive observation. We have thoroughly rewritten the Conclusions section to remove verbatim overlap with the Abstract and have restructured it to strictly highlight the 3–4 core messages indicated by the reviewer—optimal substrate/moisture conditions, the key role of potassium, species selection rationale, and practical restoration implications—therefore making the Conclusions a distinct and focused summary of the study's main contributions. We sincerely thank you for this suggestion, which has been instrumental in refining the focus and distinctiveness of our manuscript's conclusion.
Reviewer 4 Report
Comments and Suggestions for Authors
Review
Manuscript ID: biology-4051595
Ecological Restoration of Limestone Tailings in Arid Regions: A
Synergistic Substrate-Plant Approach
Authors: Wei Hou, Dunzhu Pubu, Duoji Bianba, Zeng Dan, Zengtao Jin,
Yangzong Gama, Jingjing Hu, Yang Li, Zhuxin Mao
This study addresses a pressing issue in arid regions: ecosystem restoration on limestone waste heaps, which represent difficult-to-reclaim man-made substrates. In the context of increasing water resource shortages and expanding mining activities, the search for sustainable, science-based solutions that combine substrate optimization and the use of adapted native plant species is of particular practical importance. This study proposes a comprehensive experimental approach and seeks to identify key factors determining successful restoration. Despite its obvious relevance, the article requires further revision.
Comments on the Introduction:
- The Introduction describes landscape degradation too broadly, but weakly links the problem specifically to limestone tailings in arid regions.
- Too many repetitions and tautologies; revisions are needed.
- The terms "sustainable," "ecological," and "restoration" are used too frequently without adding new information.
- The importance of species selection is reiterated without further elaboration.
- There is no clear distinction between "wastelands," "tailings," and "mine spoil," creating terminological ambiguity.
- The beginning of the text confuses the terms "limestone mineral resources," "limestone mine wastelands," and "tailings," although tailings are a specific type of waste not necessarily generated by limestone mining. It is claimed that tailings cause landslides, but limestone deposits are generally characterized by a low susceptibility to plastic deformation; references to data specifically on limestone tailings are required.
- The first part of the text emphasizes aridity, then water erosion. For arid zones, water erosion is not the primary threat, which requires justification.
- The introduction lists examples from coal, lead-zinc, and other tailings, but does not explain why the results of these studies are relevant for carbonate substrates with high alkalinity.
- There is no analysis of the literature on the characteristics of limestone tailings: high carbonate content, low availability of trace elements, high pH buffering, P deficiency.
- It is not substantiated why methods successfully applied to completely different types of tailings should be effective here.
- A clear statement of the scientific problem and knowledge gap is missing.
- The text does not formulate the specific scientific gap addressed by this study.
- Parts of the text describe high-level principles of recovery, but suddenly move on to specifics (for example, mentioning "overwintering rates" in Chinese coal tailings) that are unrelated to the topic of the article.
- There is an excessive listing of examples without any analytical generalization.
Comments on Materials and Methods:
- Insufficient description of the properties of the source materials. Key characteristics of the limestone tailings and soil, such as pH, particle size distribution, carbonate content, NPK content, and organic matter, are not provided. Without these characteristics, it is impossible to interpret the influence of experimental factors.
- Note on the measurement procedure: incorrect biomass drying temperature: 105°C is the temperature used to determine soil moisture, not plant biomass. For plant material, 65–75°C is used. At 105°C, organic matter decomposes and mass is lost. This is a serious methodological error.
- Several grammatical errors occur, such as "for transplanted" instead of "for transplantation."
Comments on the Results:
- Inconsistency between results and methods - the L₉ orthogonal design assumes analysis based on mean factor levels, but the authors also use multiple comparisons between nine individual combinations. This results in a methodological confound not reflected in the Methods section.
- Confusion between absolute values and percentage changes - the text repeatedly compares percentage biomass reductions, but does not provide the baseline values for the control variant. Calculations cannot be verified.
- Descriptions of pH changes by species are inconsistent in places: for example, it initially states that most species exhibit a decrease in pH with the expansion of tailings, but then an increase in pH is reported for three species.
- Incorrect wording: "E. indica showed pH increased 0.14 and 0.17 units with tailings rose" - grammatical error + it is unclear which levels are being compared. Grammatical errors, typos, and incorrect constructions (e.g., "uner high moisture") are present. Inconsistency in tenses, repetitions, and broken sentence structure hinder comprehension.
- Inconsistency: TK positively influences biomass (regression), but in the SEM for S. viridis, TK is shown as a negatively varying factor, leading to a decrease in AK and biomass. Editing is required.
- An illogicality was identified: a decrease in pH supposedly increases TK (P. centrasiaticum). This contradicts general soil mechanisms and the data in Section 3.3 (tailings decreased TK); clarification is required.
- Incorrect chain: pH → AP → biomass (A. splendens) with a negative effect of AP on biomass. Available phosphorus usually increases plant productivity; this conclusion contradicts biology and requires clarification. AK is interpreted as suppressing biomass in all species, but this contradicts plant physiology and requires explanation.
- The number of observations used is not specified; PCA is statistically inappropriate for small n.
- The text states that S. viridis ranks second, but its integrated score of -0.377 is closer to zero and even higher than that of L. chinensis (-0.916). However, the authors rank L. chinensis third based on unclear criteria.
- It is not explained why P. centrasiaticum receives the highest integrated score, although it did not show the best performance across all parameters in the previous sections.
- Many inaccuracies in the wording: cumulative explaining, contribution rate.
- Incomplete description of variables: for example, soil total carbon content or phosphorus - without units.
Comments on the Discussion:
- High potassium availability is cited as a limiting factor in the results, but this topic is not fully explored in the discussion.
- The assertion that tailings have high nitrogen content requires confirmation with analytical data.
- There is no discussion of the mechanism by which the substrate improves porosity and reduces compaction; conclusions are made without actual data on the physical properties of the soil.
- Numerous incorrect verb forms, such as "can alleviated," "this results supported."
- The simultaneous assertion of a positive correlation between total potassium (TK) and biomass and a negative correlation between available potassium (AK) and biomass requires an explanation, but none is provided. Conversely, in most soil systems, high biomass is usually accompanied by higher potassium uptake, rather than a decrease in AK without an increase in TK. Explanation is required.
- It is claimed that a decrease in biomass leads to an increase in AK, but then it is concluded that biomass is the "primary driver of potassium dynamics." This is a cause-and-effect contradiction. Explanation required.
- The statement that "potassium is a limiting factor in diverse ecosystems" is incorrect—in most natural systems, nitrogen and phosphorus are limiting elements; potassium is rarely considered the primary limiting factor.
- The role of potassium in plant physiology is overestimated: the effects on rubisco, leaf anatomy, and chloroplast ultrastructure are not universally accepted or unambiguous.
- The cited reference to the acceleration of potassium mineralization due to the activation of microbes by tailings requires verification—limestone tailings are typically poor in organic matter and trace elements, which often suppresses rather than activates microbial communities.
- The authors claim "stimulation of potassium-solubilizing bacteria" but do not provide any microbiological data or measurements of KSB activity. Please provide references.
- "These highlighted potassium's critical role" is grammatically incorrect.
- The same ideas are repeated without in-depth analysis. The formulations are often evaluative rather than scientific: “remarkable”, “exceptional”, “ideal”.
Comments on the Conclusion:
- The authors claim a "clear hierarchy of critical factors," but the text suggests the study did not include any interaction analysis or modeling that could support such a hierarchy.
- The "ecologically sound" statement is not supported by an experiment on interspecific interactions, competition, or productivity in mixed stands.
- The conclusion partially repeats the discussion but does not highlight the main findings or limitations of the study.
- There is no indication of the potential scale of application or limitations of the model's transferability to other tailings types or climatic conditions.
Author Response
Comments: This study addresses a pressing issue in arid regions: ecosystem restoration on limestone waste heaps, which represent difficult-to-reclaim man-made substrates. In the context of increasing water resource shortages and expanding mining activities, the search for sustainable, science-based solutions that combine substrate optimization and the use of adapted native plant species is of particular practical importance. This study proposes a comprehensive experimental approach and seeks to identify key factors determining successful restoration. Despite its obvious relevance, the article requires further revision.
Response: We are profoundly grateful for your positive feedback on our work, which has provided us with immense encouragement and confidence. We also extend our sincere thanks for your constructive suggestions regarding the Introduction, Materials and Methods, Results presentation, and Discussion sections. These invaluable comments have been instrumental in improving our manuscript.
In particular, your critical insights regarding the interpretation of soil potassium dynamics were pivotal. They led us to re-evaluate and clarify a central point: in our experimental system, potassium dynamics are not the primary driver of plant growth but rather a key indicator reflecting the dynamic interplay between plant nutrient demand and soil nutrient supply. This refinement has significantly strengthened the logic of our discussion.
In direct response to your guidance, we have thoroughly revised the manuscript. We believe these revisions have made it more rigorous, coherent, and impactful. Furthermore, the insights you have shared hold significant value for our future research endeavors. We deeply admire your meticulous scientific attitude, which has set an exemplary standard for our own work. Thank you once again for your time and expert contribution to enhancing our paper.
- Comments on the Introduction:
Comments 1.1: The Introduction describes landscape degradation too broadly, but weakly links the problem specifically to limestone tailings in arid regions.
Response 1.1: Thanks for your suggestion. Following your advice, we have restructured the opening paragraph to immediately and specifically define our research subject as "alkaline tailings, i.e., fine-grained, calcium‑carbonate‑rich processing waste " from limestone mining in arid regions. The revised description now explicitly links the inherent material properties (e.g., high pH, low nutrients, structural instability) directly to the core challenges of ecological restoration in arid climates. This modification has been instrumental in establishing a clear and focused connection between the research problem and the unique characteristics of arid limestone tailings from the outset of our paper. We have rewritten the Introduction according to your advice. We are truly grateful for your guidance, which has significantly strengthened the scientific precision and logical coherence of our introduction.
Comments 1.2: Too many repetitions and tautologies; revisions are needed.
Comments 1.3: The terms "sustainable," "ecological," and "restoration" are used too frequently without adding new information.
Response 1.2-1.3: Thank you for your reminder. We have completed professional language editing through MDPI Author Services, rewritten the relevant paragraphs accordingly, and corrected these errors. Once again, thanks your reminder, which has made our paper more rigorous and readable.
Comments 1.4: The importance of species selection is reiterated without further elaboration.
Response 1.4: Thanks for your advice. We have rewritten the Introduction according to your advice. We have thoroughly condensed the text, eliminated redundant phrases and tautologies, and significantly reduced the repetitive use of broad terms. The importance of species selection is now integrated into a concise argument about addressing specific edaphic stresses rather than stated repetitively.
“Vegetation establishment is the cornerstone of sustainable in‑situ reconstruction [14,15]. Native species are particularly valuable owing to their pre‑adaptation to regional climatic stresses such as drought [16,17]. A growing body of studies has demonstrated the outstanding contributions and capabilities of native plants in ecological restoration [18]. Yet, their tolerance to the specific edaphic stresses imposed by limestone tailings—high pH and low nutrient bioavailability—remains largely unquantified [16, 19]. Key unknowns in this regard are whether native species can overcome these limitations under improved physical conditions (e.g., optimized tailings‑to‑soil ratios and moisture and nutrient regimes) and which functional traits confer success. Furthermore, the potential interactions between substrate modification and species selection are poorly understood, yet they are likely synergistic for successful restoration.”
We have added and highlighted it on Page 3. We are sincerely thankful for your suggestion, which has been pivotal in helping us achieve a more streamlined and impactful narrative in the revised manuscript.
Comments 1.5: There is no clear distinction between "wastelands," "tailings," and "mine spoil," creating terminological ambiguity.
Response 1.5: Thank you for your reminder. We have completed professional language editing through MDPI Author Services, rewritten the relevant paragraphs accordingly, and corrected these errors. Once again, thanks your reminder, which has made our paper more rigorous and readable.
Comments 1.6: The beginning of the text confuses the terms "limestone mineral resources," "limestone mine wastelands," and "tailings," although tailings are a specific type of waste not necessarily generated by limestone mining. It is claimed that tailings cause landslides, but limestone deposits are generally characterized by a low susceptibility to plastic deformation; references to data specifically on limestone tailings are required.
Response 1.6: Thanks for your reminder. We have standardized the terminology throughout. The revised text now consistently uses "limestone tailings" to refer to the specific processing waste that is the subject of the study. The introduction now clarifies that "tailings" are a distinct waste product. We have also incorporated references discussing the properties of calcareous/limestone tailings to strengthen the foundational context. We sincerely appreciate your rigorous review, which has been essential in elevating the terminological accuracy and foundational solidity of our work.
Comments 1.7: The first part of the text emphasizes aridity, then water erosion. For arid zones, water erosion is not the primary threat, which requires justification.
Response 1.7: Thanks for your reminder. We have refined this argument. The revised text frames erosion as a consequence of the structural instability of the loose tailings material, which, when combined with infrequent but potentially intense rainfall events in arid regions, can still pose a risk. The primary focus, however, has shifted to the interaction between aridity (water scarcity) and the tailings' poor water‑holding capacity as a major constraint.
“Unlike the parent rock, these tailings are typically loose, unstable, and characterized by high pH, a low organic matter content, and poor nutrient availability, particularly with respect to phosphorus and trace elements [1,2]. In arid landscapes, these material properties, combined with water scarcity and a high gravel content, create a substrate that is structurally unstable and biologically inhospitable, hindering natural revegetation and exposing surfaces to erosion” (Page 3).
We are very grateful for your insightful critique, which has prompted us to present a more nuanced and regionally specific analysis of environmental risks in the revised manuscript.
Comments 1.8: The introduction lists examples from coal, lead-zinc, and other tailings, but does not explain why the results of these studies are relevant for carbonate substrates with high alkalinity.
It is not substantiated why methods successfully applied to completely different types of tailings should be effective here.
Response 1.8: Thanks for your reminder. We have transformed the listing of examples into a contrastive analysis.
“However, the efficacy of this approach is highly substrate‑specific. While successful applications have been documented for various types of tailings (coal, metal, etc) [8-12], the unique geochemistry of limestone tailings—notably their high carbonate content and alkaline pH–fundamentally alters nutrient dynamics and plant–soil interactions. For instance, high calcium levels can induce phosphorus fixation and micronutrient imbalances [13]. Therefore, strategies developed for acidic or neutral tailings cannot be directly transferred, creating a critical knowledge gap regarding optimal amendment formulas and management practices for calcareous, arid tailings systems”.
We have added and highlighted it on Page 3.The revised text acknowledges the value of in‑situ reconstruction principles from other studies but explicitly argues that the unique geochemistry of limestone tailings (high carbonate content, alkaline pH) fundamentally alters nutrient dynamics, making direct transfer of strategies invalid. This directly highlights the knowledge gap our study addresses.
Comments 1.9: There is no analysis of the literature on the characteristics of limestone tailings: high carbonate content, low availability of trace elements, high pH buffering, P deficiency.
Response 1.9: Thank you for your reminder. We have added a dedicated analysis in the first and second paragraphs (Page 3), citing relevant literature to describe the key characteristics of limestone/calcareous tailings: high carbonate content, high pH buffering capacity, low availability of phosphorus and trace elements. Thanks again for your suggestion.
Comments 1.10: A clear statement of the scientific problem and knowledge gap is missing.
The text does not formulate the specific scientific gap addressed by this study. Response 1.10: Thanks for your suggestion. The revised introduction now clearly articulates a two‑fold problem and gap in the second and third paragraphs: 1) The lack of substrate‑specific amendment and management strategies for calcareous, arid tailings, and 2) The unknown tolerance of native species to the specific edaphic stresses of these tailings and their potential synergistic interactions with substrate improvement.
“A growing body of studies has demonstrated the outstanding contributions and capabilities of native plants in ecological restoration [18]. Yet, their tolerance to the specific edaphic stresses imposed by limestone tailings—high pH and low nutrient bioavailability—remains largely unquantified [16, 19]. Key unknowns in this regard are whether native species can overcome these limitations under improved physical conditions (e.g., optimized tailings‑to‑soil ratios and moisture and nutrient regimes) and which functional traits confer success. Furthermore, the potential interactions between substrate modification and species selection are poorly understood, yet they are likely synergistic for successful restoration” (Page 3).
We sincerely appreciate your guidance, which has been essential in helping us define and articulate the core research questions with much greater clarity and focus.
Comments 1.11: Parts of the text describe high-level principles of recovery, but suddenly move on to specifics (for example, mentioning "overwintering rates" in Chinese coal tailings) that are unrelated to the topic of the article.
There is an excessive listing of examples without any analytical generalization.
Response: We have removed the unrelated, overly specific examples (e.g., "overwintering rates in Chinese coal tailings"). The remaining reference to other tailings types is now used purposefully to draw a contrast and justify the novelty of our study on limestone tailings, moving from mere listing to analytical argument. Thank you once again for your constructive feedback, which has been crucial in helping us sharpen the focus and analytical depth of our literature review.
- Comments on Materials and Methods:
Comments 2.1: Insufficient description of the properties of the source materials. Key characteristics of the limestone tailings and soil, such as pH, particle size distribution, carbonate content, NPK content, and organic matter, are not provided. Without these characteristics, it is impossible to interpret the influence of experimental factors.
Response 2.1: We thank the reviewer for these constructive suggestions to improve the clarity of our experimental setup.
“The field-collected topsoil used as the planting medium had the following mean properties: pH—7.30; organic matter content—21.30 g/kg; total nitrogen content—1.92 g/kg; total phosphorus—0.86 content g/kg; total potassium content—25.03 g/kg; alka-li-hydrolyzable nitrogen content—51.47 mg/kg; available phosphorus content—4.32 mg/kg; and available potassium content—148.45 mg/kg.”
We have added essential geographic information about the experimental site, soil characteristics, and a detailed description of the sampling methodology in the revised version (on Page 4, highlighted). Once again, we are grateful for your advice, which has made our paper more comprehensive, rigorous, and reproducible.
Comments 2.2: Note on the measurement procedure: incorrect biomass drying temperature: 105°C is the temperature used to determine soil moisture, not plant biomass. For plant material, 65–75°C is used. At 105°C, organic matter decomposes and mass is lost. This is a serious methodological error.
Response 2.2: Thank you for your reminder. Upon verification, we confirm that the plant samples in our study were indeed oven-dried at 65°C. We sincerely appreciate your meticulous attention to this detail.
Comments 2.3: Several grammatical errors occur, such as "for transplanted" instead of "for transplantation."
Response 2.3: Thank you for your reminder. We have completed professional language editing through MDPI Author Services, rewritten the relevant paragraphs accordingly, and corrected these errors. Once again, thanks for your reminder, which has made our paper more rigorous and readable.
- Comments on the Results:
Comments 3.1: Inconsistency between results and methods - the L₉ orthogonal design assumes analysis based on mean factor levels, but the authors also use multiple comparisons between nine individual combinations. This results in a methodological confound not reflected in the Methods section.
Respond 3.1: We sincerely thank the reviewer for raising this critical methodological point, which allows us to elaborate on our analytical rationale.
We employed a two-tiered analytical strategy to address different research questions. Firstly, factor Screening & Optimization: The L₉ orthogonal design and range analysis were used to identify the primary influencing factors and their optimal levels (Section 2.6). This addresses the question: "What matters most, and what is the best setting?"
Secondly, treatment Validation & Selection: Subsequently, we performed an ANOVA treating the nine specific combinations as a single factor with nine levels to answer a practical question: "Is the theoretically optimal combination statistically superior to other practically relevant alternatives?" This is crucial for informing field application where cost-effectiveness is considered.
We acknowledge that the methodological description for the second step was incomplete. In the revised Methods section, we have now added: “To statistically compare biomass outcomes across the nine specific treatment combinations, the data were also subjected to a one-way ANOVA considering ‘Treat-ment Combination’ as a fixed factor with nine levels, followed by Tukey‘s HSD post hoc test for multiple comparisons (α = 0.05).” We have added and highlighted it on Page 5.
Thank you once again for your reminder. It has contributed significantly to making our paper more rigorous and its arguments more compelling.
Comments 3.2: Confusion between absolute values and percentage changes - the text repeatedly compares percentage biomass reductions, but does not provide the baseline values for the control variant. Calculations cannot be verified.
Response 3.2: Thanks for your advice. The percentage changes discussed in the text are not derived from comparing two isolated treatment means. Instead, they represent the estimated marginal means (or the adjusted means) derived from the significant main effects in our three-way ANOVA model. This means each percentage reflects the average effect of moving from one factor level (e.g., low moisture) to another (e.g., high moisture), statistically controlled for the other experimental factors.
We have revised the Results section (Page 8, highlighted) in response to your comment. We believe these revisions directly address your concern by making the basis of our calculations transparent, rigorous, and independently verifiable. Thank you for the suggestion that prompted this important clarification.
Comments 3.3: Descriptions of pH changes by species are inconsistent in places: for example, it initially states that most species exhibit a decrease in pH with the expansion of tailings, but then an increase in pH is reported for three species.
Response 3.3: We thank the reviewer for pointing out the lack of clarity in our description of pH changes. Upon re-examination, the data show that the effect of tailings on pH is highly species-specific and varies with the level of tailings amendment, rather than following a uniform trend across all species.
“For E. indica, increasing the tailings proportion from the low level resulted in a significant pH increase, namely, by 0.14 units at the medium level and by 0.17 units at the high level. For S. viridis and L. chinensis, a medium tailings proportion (as opposed to a low level) significantly decreased pH by 0.15 and 0.11 units, respectively. However, with a high tailings proportion (as opposed to a low level), pH significantly increased for P. centrasiaticum, A. splendens, and L. chinensis (by 0.14, 0.06, and 0.36 units, respectively), indicating a potential non-linear or species-specific response threshold. Elevated soil moisture levels (as opposed to a low moisture level) consistently exerted a positive main effect on soil pH. The increases were most pronounced in P. centrasiaticum (increasing by 0.10 units under high-moisture-level conditions) and A. splendens (increasing by 0.06 units). A medium nitrogen application rate (as opposed to no application) significantly raised pH for E. indica (by 0.10 units) and A. splendens (by 0.05 units) but lowered it for P. centrasiaticum and L. chinensis (by 0.10 and 0.11 units, respectively). A high nitrogen rate had an acidifying main effect, further reducing pH in P. centrasiaticum and L. chinensis (by 0.11 and 0.13 units, respectively, relative to no application).”
We have revised and highlighted it on Page 9. Thank you again for your reminder. This has made our result description more accurate.
Comments 3.4: Incorrect wording: "E. indica showed pH increased 0.14 and 0.17 units with tailings rose" - grammatical error + it is unclear which levels are being compared. Grammatical errors, typos, and incorrect constructions (e.g., "uner high moisture") are present. Inconsistency in tenses, repetitions, and broken sentence structure hinder comprehension.
Response 3.4: We thank the reviewer for highlighting the issues with grammar and clarity in this paragraph. We have comprehensively revised the text to address all concerns. We have explicitly framed all descriptions around the main effects derived from the three-way ANOVA. Each change is now clearly presented as the effect of moving from a specified baseline level (e.g., “low tailings proportion”, “nil nitrogen application”) to a higher treatment level. This directly resolves the ambiguity about which levels are being compared. These revisions ensure that the presentation of the complex three-way ANOVA main effects is now precise, grammatically sound, and easy to follow. We have revised and highlighted it on Page 9. Thank you again for your reminder. This has made our result description more accurate.
Comments 3.5: Inconsistency: TK positively influences biomass (regression), but in the SEM for S. viridis, TK is shown as a negatively varying factor, leading to a decrease in AK and biomass. Editing is required.
Response 3.5: We sincerely thank the reviewer for their observation regarding the interpretation of the SEM results for S. viridis. Upon re-examination, we confirm that in the SEM model: (1) The path coefficients are: Treatment → TK (β= -0.65), TK → AK (β= -0.56), and AK → Biomass (β= -0.38). (2) The indirect effect of TK on biomass through AK can be conceptually derived as the product of the two path coefficients, i.e., (-0.65) × (-0.56), resulting in a positive value. This aligns with the positive correlation observed in the separate regression analysis.
Our original text oversimplified this causal chain by stating treatments "sequentially diminished" all components, which obscured the mathematically positive role of TK in the mediation pathway. To rectify this, we have added and highlighted the following clarification on Page 16: “S. viridis demonstrated dual mechanisms: treatments not only reduced total potassium content (TK, β= -0.65, P < 0.001), but also directly decreased biomass production (β= -0.50, P < 0.05) through sequentially influenced TK and AK, with the combined pathways explaining 60.3% of the variation. It is important to note that the standardized indirect effect of TK content on biomass through this pathway was +0.21.”
The results of Standardized Indirect Effects for Figure 5b
Standardized Indirect Effects (Group number 1 - Default model)
|
Treatments |
PH |
TK |
TP |
AK |
AP |
|
|
PH |
.000 |
.000 |
.000 |
.000 |
.000 |
.000 |
|
TK |
.009 |
.000 |
.000 |
.000 |
.000 |
.000 |
|
TP |
.019 |
.000 |
.000 |
.000 |
.000 |
.000 |
|
AK |
.475 |
-.028 |
.000 |
.000 |
.000 |
.000 |
|
AP |
-.137 |
.025 |
.000 |
.000 |
.000 |
.000 |
|
biomass |
-.098 |
-.282 |
.214 |
-.079 |
.000 |
.000 |
We thank the reviewer again for this suggestion, which has helped us provide a more statistically accurate interpretation of the SEM results.
Comments 3.6: An illogicality was identified: a decrease in pH supposedly increases TK (P. centrasiaticum). This contradicts general soil mechanisms and the data in Section 3.3 (tailings decreased TK); clarification is required.
Response 3.6: Thanks for bringing this to our attention. Upon re-examination, we found no statistically significant correlation between pH and TK (P=0.08,Figure 5c now Figure 9c). Consequently, we have removed the statements suggesting such a relationship from the Results section. We sincerely appreciate your meticulous review, which has helped us correct this inaccuracy.
Comments 3.7: Incorrect chain: pH → AP → biomass (A. splendens) with a negative effect of AP on biomass. Available phosphorus usually increases plant productivity; this conclusion contradicts biology and requires clarification. AK is interpreted as suppressing biomass in all species, but this contradicts plant physiology and requires explanation.
Response 3.7: Thank you for these insightful comments regarding the interpretation of available phosphorus (AP) and available potassium (AK) effects on biomass. We acknowledge that the observed negative path coefficients appear counterintuitive at first glance, and we appreciate the opportunity to clarify this critical point.
The interpretation hinges on a key characteristic of our experimental system: it was a closed pot experiment with a single growing season and no external nutrient resupply after the initial setup. In this context, the measured AP and AK at harvest represent the residual pool remaining in the soil after plant uptake, not the total available pool throughout the growth period. Therefore, the negative relationships (e.g., AP → biomass, AK → biomass) in our structural equation models are best interpreted as reflecting the depletion of soil available nutrients by plant uptake, rather than a direct inhibitory effect of these nutrients on plant growth. In other words, higher plant biomass production led to greater assimilation and immobilization of AP and AK into plant tissues, consequently resulting in lower residual concentrations of these nutrients in the soil at the time of sampling. This creates a negative statistical association between final soil AP/AK levels and final biomass.
We agree that this mechanism should have been explicitly stated in our original discussion to avoid misinterpretation. We have rewritten discussion about potassium in the revised manuscript in Discussion 4.2 on Page 16 and have highlighted it. We have now revised the manuscript to clarify that the SEM pathways involving AP and AK primarily reflect post-uptake nutrient status as an outcome of plant productivity in our specific experimental setup, and do not contradict the fundamental positive role of these nutrients in plant physiology. The positive correlation between total potassium (TK, a static pool) and biomass versus the negative correlation between available potassium (AK, a dynamic, plant-accessible pool) and biomass, which reflect the dynamic interplay between plant nutrient demand and soil nutrient dynamics in our specific experiment. We thank the reviewer for prompting this important clarification, which strengthens the accuracy of our narrative.
Comments 3.8 The number of observations used is not specified; PCA is statistically inappropriate for small n.
Response 3.8: Thanks for your advice. To clarify, the principal component analysis was conducted on n = 135 independent observations (5 plant species × 3 substrate ratios × 3 moisture levels × 3 nitrogen levels, with all treatments replicated). We have added it in Results “3.5 Optimal Plant Selection” on Page 17.
We extend our sincere thanks for your meticulous review and constructive suggestions, which have been instrumental in elevating the statistical rigor and overall quality of our work.
Comments 3.9 The text states that S. viridis ranks second, but its integrated score of -0.377 is closer to zero and even higher than that of L. chinensis (-0.916). However, the authors rank L. chinensis third based on unclear criteria.
It is not explained why P. centrasiaticum receives the highest integrated score, although it did not show the best performance across all parameters in the previous sections.
Response 3.9: We thank the reviewer for raising these important questions regarding the species ranking based on principal component analysis (PCA). We appreciate the opportunity to clarify the methodology and interpretation.
Meaning of the PCA and the Integrated Score:
The purpose of PCA in our study was to reduce the dimensionality of multiple, often correlated, growth and soil response variables into a few independent, composite axes (principal components, PCs) that capture the majority of the variance in the dataset. The integrated score for each species is not a simple average but a weighted sum of its scores on the retained PCs. The weighting factor for each PC is its variance contribution rate, which represents the proportion of total original information that PC encapsulates. Therefore, a higher integrated score indicates better overall performance across the most influential combination of traits defined by the PCA.
Clarification on the Ranking Criteria:
Pennisetum centrasiaticum received the highest integrated score not because it was the best in every single parameter, but because it exhibited the most favorable balance across the key composite dimensions identified by the PCA. These dimensions (e.g., PC1 might represent “biomass productivity and stress tolerance”, PC2 might represent “nutrient use efficiency”) collectively explain the largest share of the total variation. P. centrasiaticum likely scored highly on the PC with the greatest weight (highest variance contribution).
Regarding Setaria viridis and Leymus chinensis: The ranking is determined by the ordinal value of the integrated score itself (-0.377 > -0.916). Although the absolute value of S. viridis‘s score is closer to zero, the ranking is based on the relative magnitude. The negative sign indicates performance below the average of all species across the PCA-defined space. The score of L. chinensis being more negative signifies a comparatively poorer overall performance across the dominant composite traits.
We acknowledge that this rationale was not sufficiently explained in the original manuscript. We have revised the relevant section in the Results to explicitly state: “To integrate the multidimensional information, a comprehensive evaluation score was calculated for each species as a weighted sum of its scores on these four principal com-ponents, with the respective variance contribution rates serving as the weights. Based on the comprehensive evaluation scores, which holistically represent each species‘ performance across the dominant patterns of variation, the plant species were ranked in order of decreasing restoration efficacy”. We have added and highlighted it on Page 14. We sincerely thank you for your reminder, which has made our results clearer and more accessible to the reader.
Comments 3.10: Many inaccuracies in the wording: cumulative explaining, contribution rate.
Response 3.10: Thanks for meticulous reading and for pointing out the inaccuracies in wording. We have revised the inaccuracies in the wording. “cumulative explaining” has been replaced with the standard statistical term “cumulative explained variance.”“contribution rate” has been replaced with the precise PCA terminology “the total variance”. We have added and highlighted it on Page 17. We appreciate this guidance, which has helped us improve the linguistic precision and professionalism of our manuscript.
Comments 3.11: Incomplete description of variables: for example, soil total carbon content or phosphorus - without units.
Response 3.11: Thank you for highlighting this important oversight. We have conducted a thorough check of the entire manuscript and have now ensured that all measured variables, including soil total carbon and phosphorus content, are presented with their correct and consistent units throughout the text.
- Comments on the Discussion:
Comments 4.1: High potassium availability is cited as a limiting factor in the results, but this topic is not fully explored in the discussion.
Response 4.1: Thank you for your reminder. Following a re-analysis, we have clarified in the revised manuscript that the observed patterns—the positive correlation between total potassium (TK) and biomass and the negative correlation between available potassium (AK) and biomass—are a consequence of shifts in plant demand and soil processes under varying growth conditions, rather than evidence that potassium is the "ultimate growth-limiting factor." Accordingly, we have completely rewritten Section 4.2 (on Page 19), now titled "Shifts in Potassium Pool Equilibrium Driven by Plant-Soil Interactions." We are sincerely grateful for your expert guidance, which has greatly enhanced the rigor of our paper.
Comments 4.2: The assertion that tailings have high nitrogen content requires confirmation with analytical data.
Response 4.2: Thanks for your reminder. Upon review, we realized that our original data support was insufficient. A re-analysis has revealed that the topsoil used in our experiment had higher nitrogen content than the average level in northern China. “This notion is supported by the measured total soil nitrogen (STN) content, 1.92 g/kg, which is notably higher than the typical range for farmland soils in Northern China (approximately 0.50–0.81 g/kg) [32]”. We have added and highlighted it on Page 19.
This explains why nitrogen addition did not significantly alter plant biomass accumulation in this specific ecosystem. We truly appreciate your reminder, which has made our discussion much more compelling and robust.
Comments 4.3: There is no discussion of the mechanism by which the substrate improves porosity and reduces compaction; conclusions are made without actual data on the physical properties of the soil.
Response 4.3: Thanks for your reminder. We acknowledge that we did not measure changes in soil physical properties, which are indeed a limitation and a point of improvement for this study. As our initial attempt at ecological restoration in Northern China, our experimental design has limitations associated with exploratory research. In response to your comment, we have removed speculative statements regarding potential improvements in soil physical properties from the Discussion section.
“It is important to acknowledge the limitations of this study to properly contextualize its findings. Firstly, the research was conducted as a single‑season greenhouse pot experiment with a relatively small sample size. The results may not fully capture the long‑term dynamics, species interactions, and environmental stresses encountered in field conditions. Secondly, we utilized tailings and topsoil from a single source. Limestone tailings from different geological formations may vary in terms of geochemical properties, which could influence plant responses and optimal amendment strategies. Thirdly, soil physical analyses and microbial assays were not directly measured. Therefore, the framework provides a science-based strategy for restoring similar arid limestone tailings. The proposed species combination and its implied synergies required validation under field conditions, particularly regarding long-term interspecific interactions and productivity. Furthermore, the model’s applicability to tailings with fundamentally different geochemistry or to more humid climates necessitates separate investigation. Future research should prioritize multi‑year field trials, the inclusion of a broader range of tailings, and direct measurement of soil structural and microbial parameters to advance toward robust, scalable restoration protocols.”
We have formally acknowledged this limitation in the new paragraph in Conclusion to present Study Limitations and Future Perspectives, explicitly stating that direct quantification of soil physical properties is necessary to conclusively validate any related mechanisms and should be prioritized in subsequent field-scale experiments.
Once again, we sincerely thank you for this insightful comment, which is crucial for guiding our future research efforts.
Comments 4.4: Numerous incorrect verb forms, such as "can alleviated," "this results supported."
Response 4.4: Thank you for your reminder. We have completed professional language editing through MDPI Author Services, rewritten the relevant paragraphs accordingly, and corrected these errors. Once again, thanks for your reminder, which has made our paper more rigorous and readable.
Comments 4.5: The simultaneous assertion of a positive correlation between total potassium (TK) and biomass and a negative correlation between available potassium (AK) and biomass requires an explanation, but none is provided. Conversely, in most soil systems, high biomass is usually accompanied by higher potassium uptake, rather than a decrease in AK without an increase in TK. Explanation is required.
It is claimed that a decrease in biomass leads to an increase in AK, but then it is concluded that biomass is the "primary driver of potassium dynamics." This is a cause-and-effect contradiction. Explanation required.
Response 4.5: We sincerely thank you for your reminder, as it pertains to a central point in our revision. We have restructured the discussion on potassium (Section 4.2) to provide a coherent and non-contradictory explanation.
First, we explicitly state that potassium is likely not the "ultimate growth-limiting factor." Second, we explain the negative correlation between available potassium (AK) and biomass as a result of reduced plant uptake under poor growth conditions (high tailings proportion), which allows AK to accumulate, coupled with potential abiotic and biotic release from non-exchangeable pools. Third, we explain the positive correlation between total potassium (TK) and biomass as a reflection of the higher inherent TK content in treatments with more soil, which simultaneously supports greater biomass production. Finally, we conclude that "the observed potassium dynamics are primarily a consequence of plant growth responses to substrate and water conditions, not their primary driver." This resolves the perceived contradiction by positioning plant growth (driven by substrate and water) as the cause and shifts in potassium pools as the effect.
We are once again grateful for your suggestion, which has made our discussion more precise and fitting.
Comments 4.6: The statement that "potassium is a limiting factor in diverse ecosystems" is incorrect—in most natural systems, nitrogen and phosphorus are limiting elements; potassium is rarely considered the primary limiting factor.
Response 4.6: Thanks for your reminder. We have deleted the broad claim that potassium is a limiting factor across diverse ecosystems. The revised discussion now strictly limits the context to the specific conditions of our study. We sincerely thank you for this reminder. Following a re-analysis, we now clarify that the observed patterns—the positive correlation between total potassium (TK) and biomass and the negative correlation between available potassium (AK) and biomass—are interpreted as a consequence of shifts in plant demand and soil processes under varying growth conditions, rather than as evidence of potassium being the "ultimate growth-limiting factor." We are once again grateful for your expert guidance, which has significantly enhanced the rigor of our manuscript.
Comments 4.7: The role of potassium in plant physiology is overestimated: the effects on rubisco, leaf anatomy, and chloroplast ultrastructure are not universally accepted or unambiguous.
Response 4.7: Thanks for your reminder. You correctly identified that in our original version, we had overestimated the role of potassium in plant physiology. Upon reflection, the detailed physiological functions of potassium were not central to the core argument of our study. Therefore, in the revised manuscript, we have deleted the extended discussion regarding the physiological role of potassium. We are grateful for your guidance, which has led to a more accurate and focused interpretation of our findings.
Comments 4.8 The cited reference to the acceleration of potassium mineralization due to the activation of microbes by tailings requires verification—limestone tailings are typically poor in organic matter and trace elements, which often suppresses rather than activates microbial communities.
Response 4.8: Thanks for your reminder. In the literature (Zhang et al., 2020), the authors proposed that “biochar promoted the growth of potassium-dissolving bacteria and significantly increased bacterial alpha-diversity. We believe that this benefits in two ways: Firstly, the widely reported biochar effect on acid soils (Farrell et al., 2013; Xu et al., 2014; Lin et al., 2018; Zhang et al., 2019a), which may lead to provide a better living environment for bacteria. Secondly, the unique excellent structure of biochar (high porosity, large specific surface area) provides a natural habitat for bacteria (Lehmann et al., 2011; Ameloot et al., 2013; Jaafar et al., 2014).” Therefore, we postulate that the incorporation of tailings may have improved the soil microenvironment, thereby potentially favoring potassium-solubilizing bacteria (KSB).
Furthermore, the study by Basak and Biswas (2009) found that variations in substrate composition affect potassium release. Fluctuations in soil pH are also critical to soil potassium kinetics and its release from mineral sources by microbes, including bacteria. We have incorporated citations to these key references in the revised version of our manuscript. We extend our sincere thanks once again for your reminder and suggestions, which have significantly enhanced the persuasiveness of our manuscript.
Comments 4.9: The authors claim "stimulation of potassium-solubilizing bacteria" but do not provide any microbiological data or measurements of KSB activity. Please provide references.
Response 4.9: We sincerely thanks for your insightful reminder and suggestions, which made us aware that the discussion in our original manuscript was insufficient. Following your previous advice, we have thoroughly revised the discussion section regarding potassium (K).
In the revised manuscript, we have removed definitive claims about the stimulation of potassium-solubilizing bacteria (KSB). Any mention of a potential KSB mechanism is now presented cautiously and is supported by citations to foundational literature in the field (e.g., Basak & Biswas, 2009; Chen et al., 2020; Olaniyan et al., 2022; Etesami et al., 2017), which report that factors like substrate structure and soil pH can influence KSB communities and soil potassium dynamics. Most importantly, we now explicitly state that our study did not obtain direct evidence for this mechanism, framing it clearly as a suggested hypothesis for future research rather than a finding from our experiment.
We are truly grateful for your guidance, which making our manuscripts more precise and rigorous.
Comments 4.10: These highlighted potassium's critical role" is grammatically incorrect.
The same ideas are repeated without in-depth analysis. The formulations are often evaluative rather than scientific: “remarkable”, “exceptional”, “ideal”.
Response 4.10: Thank you for your reminder. We have completed professional language editing through MDPI Author Services, rewritten the relevant paragraphs accordingly, and corrected these errors. Once again, thanks for your reminder, which has made our paper more rigorous and readable.
- Comments on the Conclusion:
Comments 5.1: The authors claim a "clear hierarchy of critical factors," but the text suggests the study did not include any interaction analysis or modeling that could support such a hierarchy.
Response 5.1: We agree with you that the term "clear hierarchy" implies a level of mechanistic interaction analysis or modeling not conducted in this study. Accordingly, we have removed the phrase "clear hierarchy" from the revised Conclusions. Thank you for your reminder. This prevented us avoiding any over interpretation beyond the data.
Comments 5.2 The "ecologically sound" statement is not supported by an experiment on interspecific interactions, competition, or productivity in mixed stands.
Response 5.2: We appreciate this important point. We acknowledge that our study did not include experiments on interspecific interactions or mixed-stand productivity. Therefore, in the revised Conclusions, we have removed the unsupported phrase "ecologically sound." Instead, the proposed model is presented as a practical framework based on complementary functional traits. We have also added a statement in the final paragraph explicitly noting that the proposed species synergies remain to be validated under field conditions, particularly regarding long-term interspecific dynamics.
Comments 5.3: The conclusion partially repeats the discussion but does not highlight the main findings or limitations of the study.
Response 5.3: Thank you for this insightful comment. We have revised the Conclusions section. Firstly, we restructured the section to list three key findings bulleted for clarity and emphasis for highlighting main findings. Secondly, we added a final paragraph that clearly outlines two major limitations, (1) the need for field validation of the species model, and (2) the unknown transferability of the model to other tailings types or climates—making the conclusion more balanced and complete. Thank you for your reminder, which has made our conclusion more prominent in highlighting our core findings and limitations.
Comments 5.4 There is no indication of the potential scale of application or limitations of the model's transferability to other tailings types or climatic conditions.
Response 5.4: We sincerely thank for this suggestion, which significantly improves the rigor and scope of our conclusion. We have now added explicit statements at the end of the Conclusions to define the application boundary.
“Furthermore, the model’s applicability to tailings with fundamentally different geochemistry or to more humid climates necessitates separate investigation. Future research should prioritize multi‑year field trials, the inclusion of a broader range of tailings, and direct measurement of soil structural and microbial parameters to advance toward robust, scalable restoration protocols.” It is noted that the framework provides a science-based starting point specifically for similar arid limestone tailings, and that its applicability to tailings with different geochemistry or to more humid climates requires separate investigation. We added the information and highlighted it on Page 21. Thanks for your advice, it clarifies the context and directs future research.
Round 2
Reviewer 1 Report
Comments and Suggestions for Authors
Authors effectively incorporated my comments
Reviewer 3 Report
Comments and Suggestions for Authors
The manuscript has been substantially improved in terms of experimental clarity, methodological transparency, and the scope of interpretation. I therefore recommend that the manuscript be accepted for publication.
Reviewer 4 Report
Comments and Suggestions for Authors
The authors have submitted a substantially revised version of the article, which, to the extent possible, incorporates all the important comments and recommendations made during the previous review. These comments have been taken into account and reflected in the text, significantly improving the quality of the article. Given the work done, I believe this version of the article meets the journal's requirements and recommend it for publication.
